# StableI2I: Spotting Unintended Changes in Image-to-Image Transition

Jiayang Li [1 2 * +]  Shuo Cao [2 3 *]  Xiaohui Li [2 4]  Zhizhen Zhang [1]  Kaiwen Zhu [2 4]  Yule Duan [1]  Yu Qiao [2]
Jian Zhang [1 5 †]  Yihao Liu [2 †]

## Abstract

In most real-world image-to-image (I2I) scenarios, existing evaluations primarily focus on instruction following and the perceptual quality or aesthetics of the generated images. However, they largely fail to assess whether the output image preserves the semantic correspondence and spatial structure of the input image. To address this limitation, we propose StableI2I, a unified and dynamic evaluation framework that explicitly measures content fidelity and pre–post consistency across a wide range of I2I tasks without requiring reference images, including image editing and image restoration. In addition, we construct StableI2I-Bench, a benchmark designed to systematically evaluate the accuracy of MLLMs on such fidelity and consistency assessment tasks. Extensive experimental results demonstrate that StableI2I provides accurate, fine-grained, and interpretable evaluations of content fidelity and consistency, with strong correlations to human subjective judgments. Our framework serves as a practical and reliable evaluation tool for diagnosing content consistency and benchmarking model performance in real-world I2I systems. The project page and source code are publicly available at https://henry-lee-real.github.io/StableI2I_Page.

## 1. Introduction

With the rapid advancement of generative models (Labs et al., 2025; Zhang et al., 2023b), current systems are in-

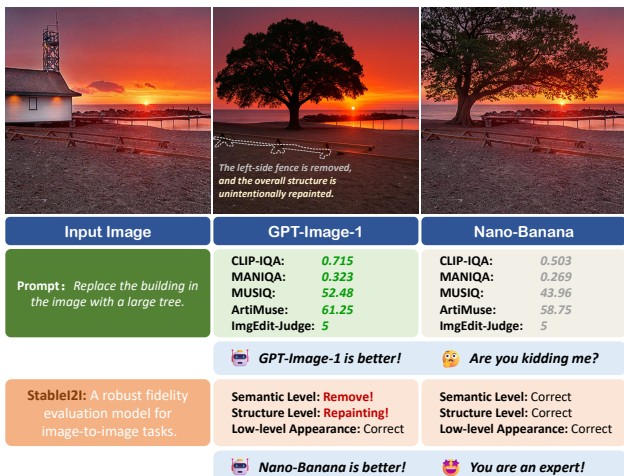

*Figure 1.* Qualitative image editing results from GPT-Image-1 and Nano-Banana, with scores from multiple evaluation metrics. CLIP-IQA (Wang et al., 2023), MANIQA (Yang et al., 2022), and MUSIQ (Ke et al., 2021) are conventional IQA metrics, while ArtiMuse (Cao et al., 2025b) is a recent IAA metric. ImgEdit-Judge (Ye et al., 2025) reports scores under the Physical & Detail Coherence dimension. In contrast, StableI2I more accurately assesses content fidelity and consistency.

creasingly capable of following user instructions and producing high-quality images. However, the inherent randomness of the sampling process often leads to substantial information drift between the generated output and the input image. Even state-of-the-art models such as Nano-Banana (Google, 2025) are affected by this issue. This phenomenon highlights the urgent need for effective methods to evaluate and calibrate content drift.

However, current I2I evaluations mainly focus on instruction following and output aesthetics (Cao et al., 2025b) or perceptual quality (Wang et al., 2023), while largely ignoring whether the output image remains faithful to the input image during editing or restoration (Fig. 1). Although images generated by GPT-Image-1 achieve higher scores under existing metrics, their texture and semantic content still exhibit unintended changes, including unnecessary repainting of the sky and sandy areas, and the disappearance of the left-side fence relative to the input image. Without explicitly assessing pre–post consistency, such inconsistencies can lead to severe consequences in high-stakes I2I applications, such as medical imaging and remote sensing. Therefore, a principled evaluation method is required to jointly consider

*Equal contribution + Work done during an internship at Shanghai Artificial Intelligence Laboratory. [1] School of Electronic and Computer Engineering, Peking University [2] Shanghai Artificial Intelligence Laboratory [3] University of Science and Technology of China [4] Shanghai Jiao Tong University [5] Guangdong Provincial Key Laboratory of Ultra High Definition Immersive Media Technology, Shenzhen Graduate School, Peking University. Correspondence to: Jian Zhang <zhangjian.sz@pku.edu.cn>, Yihao Liu <liuyihao@pjlab.org.cn>.

*Proceedings of the 43rd International Conference on Machine Learning*, Seoul, South Korea. PMLR 306, 2026. Copyright 2026 by the author(s).

the input image, output image, and processing instruction to assess content fidelity before and after transformation.

For image editing tasks, a commonly adopted strategy (Ryu et al., 2025) is to use a mask to separate the edited region and then compare the remaining areas for consistency. However, valid edits often give rise to necessary global variations, such as changes in illumination, shadows, or other secondary effects that are causally induced by the edit itself. For example, in the output images of Fig. 1, after the object is replaced with a tree, the shadow cast beneath it is a reasonable and physically plausible outcome. In such cases, rigid mask-based separation becomes inappropriate and can easily lead to erroneous judgments. Moreover, mask-based methods are not applicable to image restoration tasks, where the entire image may be altered. Consequently, an effective I2I evaluation framework must be capable of understanding the editing instruction, interpreting image content, and dynamically producing analysis results conditioned on both.

Recent studies (Liu et al., 2025) have also recognized this limitation and attempted to address it by leveraging prompt engineering to query powerful proprietary MLLMs for consistency judgments. Although current closed-source MLLMs exhibit strong semantic-level image understanding capabilities, they remain insensitive to fine-grained pixel-level and structural information (Cao et al., 2025a). As a result, such evaluation methods often produce cases where semantic content appears consistent while pixel-level content is misaligned. As shown in Fig. 1, ImgEdit-Judge assigns an incorrect score under the *Physical & Detail Coherence* dimension, failing to detect substantial content repainting. This deficiency arises because ImgEdit-Judge is distilled from the closed-source GPT-4o model (Achiam et al., 2023) and lacks explicit sensitivity to pixel-level structure.

Motivated by these observations, we propose **StableI2I**, a fidelity-oriented I2I evaluation model that jointly considers semantic and pixel-level consistency. By integrating these dimensions, StableI2I better judges semantic content and pixel-level details between the input and output images.

StableI2I adapts to different I2I tasks by conditioning on the input instruction and selectively attending to regions and attributes that must remain consistent. We further define three complementary fidelity dimensions: Structure Level, Semantic Level, and Low-level Appearance. In addition, we introduce **StableI2I-Bench**, a benchmark with formatted question–answer pairs for systematically evaluating modern MLLMs on I2I fidelity assessment across these three dimensions, reflecting both high-level semantic reasoning and low-level visual perception. We also propose an *error-amplification* data construction pipeline to mitigate the long-tail distribution of subtle consistency violations. In summary, our main contributions are as follows:

- We propose **StableI2I**, a fidelity-oriented evaluation

model for I2I tasks that jointly captures semantic-level and pixel-level consistency.

- We introduce **StableI2I-Bench**, a benchmark designed to assess models' integrated high-level and low-level visual reasoning abilities for fidelity evaluation.

- We develop a multi-stage, multi-task data construction pipeline that enhances data diversity and improves the robustness of model capabilities.

## 2. Related Works

### 2.1. Quality Assessment for I2I Transition

Quality assessment models for natural image transition are conventionally classified into Full-Reference (FR) and No-Reference (NR) paradigms (Wang et al., 2004; Zhang et al., 2018; Heusel et al., 2017; Wang et al., 2023; Hessel et al., 2021; Wu et al., 2023; You et al., 2025; Cao et al., 2025b;a). While FR metrics (Prashnani et al., 2018; Ding et al., 2020) rely on ground truths that are often unavailable, standard NR methods predominantly evaluate absolute aesthetics or perceptual quality (Wu et al., 2023; Cao et al., 2025a), failing to capture the semantic consistency with the source input that is essential for image editing. This limitation motivates a source-conditioned evaluation paradigm that explicitly accounts for content fidelity and structural preservation in the absence of ground truth.

### 2.2. MLLM-based I2I Transition Assessment

Evaluating I2I transition requires a multi-dimensional perspective that encompasses semantic consistency and aesthetic quality, yet this critical domain remains largely underexplored. Prior works (Ye et al., 2025; Liu et al., 2025; Xu et al., 2023; Cvejic et al., 2025) primarily rely on general-purpose MLLMs, either through prompt engineering as in MagicBrush (Zhang et al., 2023a) and CompBench (Jia et al., 2025), or via distillation methods such as ImgEdit (Ye et al., 2025), which trains a judge using GPT-4o (Achiam et al., 2023) priors without specific adaptation for I2I transition. Consequently, these approaches are predominantly coarse-grained and biased toward high-level semantic consistency, often failing to capture low-level pixel-wise variations or provide professional-grade diagnostic depth. These limitations highlight the need for a fidelity-centric and instruction-aware evaluation framework that jointly considers both semantic and perceptual consistency.

## 3. From Data to Benchmark and Training

### 3.1. Data Construction Pipeline

I2I tasks can be broadly categorized into two types: high-level semantic editing and low-level image restoration. Because these two task types emphasize different objectives,

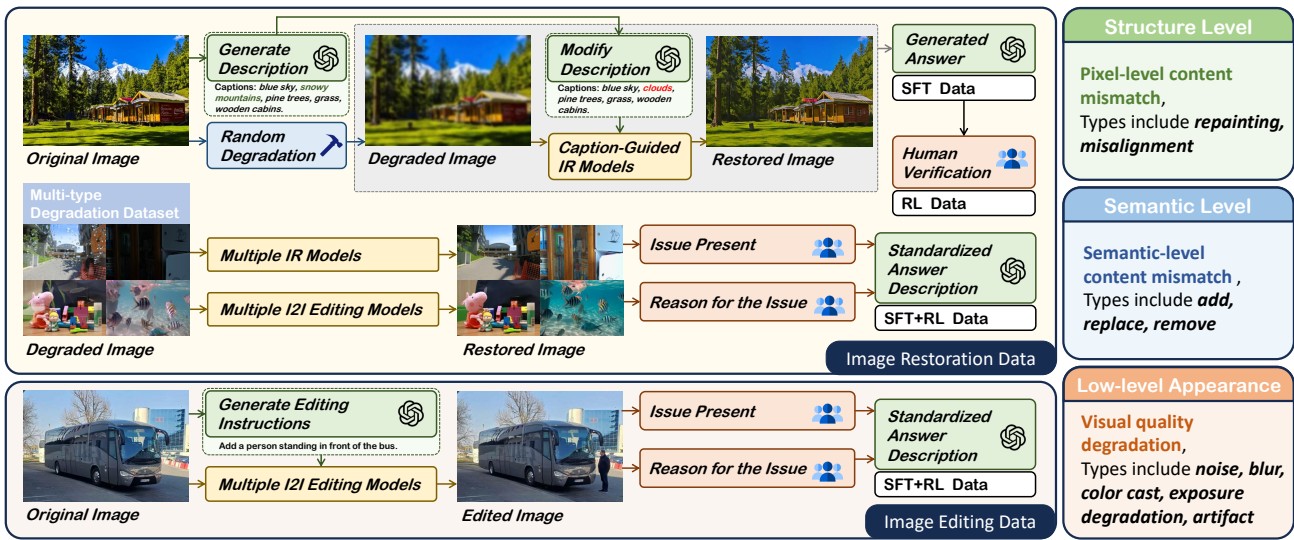

Figure 2. The data construction pipeline consists of two parts: image editing data and image restoration data. For restoration, we include both enhancement-based synthetic data and annotations on degraded outputs from real models; for editing, all data are annotated based on outputs from real models. "SFT" and "RL" denote the training stage where the data is used. The degradation types in the Multi-type Degradation Dataset are detailed in Appendix A.1. All data are annotated along the three dimensions shown on the right.

most existing models tend to focus primarily on either high-level semantics or low-level perceptual quality, while paying insufficient attention to the other, which often leads to fidelity issues. **For image editing**, models focus on preserving and modifying object-level content, which makes it difficult to maintain low-level texture details. As a result, many existing models exhibit unintended content repainting and pixel-level mismatches in regions that should remain unchanged, even though object-level semantics are preserved. **For image restoration**, models may not truly understand what object should be restored, i.e., they lack sufficient semantic capability, which leads to semantic drift in the restored content.

To address these issues, we design an error-amplification data generation pipeline together with a corresponding annotation pipeline. As shown in Fig. 2, for the image restoration task, we first apply random degradations to collected natural images. We then use GPT-5 to extract faithful content descriptions of the original images and introduce controlled semantic perturbations to deliberately alter and corrupt these descriptions. The corrupted descriptions are subsequently used to guide a text-guided image restoration model for restoration. In this way, the restoration model is guided by incorrect semantic information and is forced to restore the low-quality image toward an incorrect semantic direction, which significantly increases the probability of generating erroneous samples. For the image editing task, since it is difficult to deliberately construct erroneous data through a deterministic pipeline, we generate diverse editing results using multiple types of editing instructions together with multiple generative models. The specific models, data sources, and dataset scales used in our data pipeline are detailed in Appendix A.1.

Based on the above I2I data, we define three categories of error types, as illustrated in Fig. 2: **Semantic Level**: whether unintended additions, deletions, or modifications occur in semantic content that should be preserved; **Structure Level**: whether the output image exhibits texture or structural misalignment relative to the input image, or unintended content repainting; **Low-level Appearance**: whether the output image exhibits low-level degradations relative to the input image, such as noise, blur, color shift, or artifacts. With these three fidelity dimensions defined, we annotate two types of data, as illustrated in Fig. 2. For the image restoration task, we adopt a semi-automatic annotation scheme: for pipeline-synthesized data, since the corrupted semantic information is known by construction, we use the GPT-5 API for first-stage automatic annotations, followed by human filtering and correction; for restoration results obtained under real-world settings, we rely on fully manual annotation to label all fidelity-related content. For all data from the image editing task, we also employ fully manual annotation to label the complete content. Details of the number of annotators and the annotator training procedure are provided in Appendix A.2.

### 3.2. StableI2I-Bench: Benchmark Definition

Most existing I2I tasks rely on prompt engineering to let closed-source models evaluate the pre–post consistency of I2I results. To assess whether existing open-source and closed-source models can truly use prompts to correctly judge consistency, we release StableI2I-Bench.

We randomly sample 1,000 human-annotated image pairs from each of the three dimensions—Semantic Level, Structure Level, and Low-level Appearance—to construct the benchmark. The benchmark adopts a formatted prompt de-

**Free-form Descriptive**

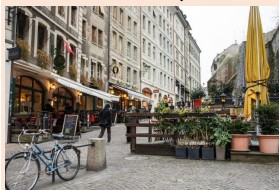

*The image captures a vibrant street scene in a European city. The street, bustling with life, is lined with buildings and shops, their mostly white and gray facades punctuated by colorful flags and signs... A blue bicycle, parked casually on the sidewalk, lends a touch of charm to the scene.*

**Binary & Type QA**

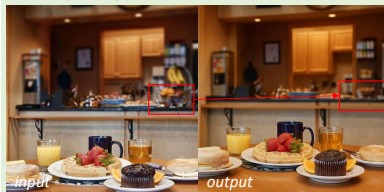

| Semantic Level |
| --- |
| No (["remove"]) |
| Structure Level |
| No (["repainting"]) |
| Low-level Appearance |
| Yes (NULL) |

I2I Instruction Prompt： *Remove the bananas in the background*

**Multiple-choice QA**

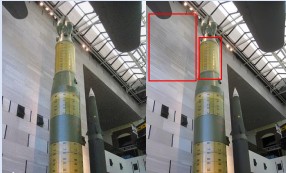

Which region or object shows an unintended visual issue in the processed image compared with the original?

A: ceiling lights
**B: missile nose cone**
**C: wall tiles**
D: overhead dark wing

I2I Instruction Prompt： *Remove noise from the image*

**Open-ended QA**

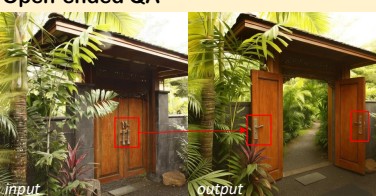

**Think:** *Check that roof, walls, plants, floor, and fixtures stay the same while allowing the door to open.*
**Problem:** *{'replace': 'Roof eaves and door handles changed in style/color compared to the original.'}*

I2I Instruction Prompt： *Make the door appear open*

*Figure 3.* An illustration of the four types of training data: Free-form Descriptive, Binary & Type QA, Multiple-choice QA, and Open-ended QA.

sign, where each prompt includes the input image, the output image, the I2I control instruction, background knowledge describing the evaluation dimension, and a specification of the required output format, together with the corresponding structured answers. The detailed prompt templates and benchmark examples are provided in Appendix A.3.

### 3.3. StableI2I-Train: Training Corpus Construction

Since StableI2I is fine-tuned on a relatively small 8B-parameter MLLM (Team, 2025), and given the limited model capacity at this scale, we adopt fixed task templates during training to ensure stable and reliable evaluation behavior. We first define two fundamental data types: **Binary & Type QA**, which produces concise and standardized evaluation outputs following Format 1, and **Open-ended QA**, which provides detailed natural-language descriptions of observed errors, with output structures corresponding to Format 2. Concrete examples of both output formats are shown in Fig. 3, and the fixed input templates for these two task types are provided in Appendix A.4.2.

```
{ "answer": Yes or No, "problem": Null
  or [Type₁, Type₂, ...] }
```

**Format 1.** Unified output format for Binary & Type QA.

```
{ "think": CoT, "problem": { "Type₁":
Detail₁, "Type₂": Detail₂, ...} }
```

**Format 2.** Unified output format for Open-ended QA.

In addition, to preserve the model's basic visual perception and descriptive abilities, we introduce a multi-task descriptive QA dataset termed **Free-form Descriptive**, as illustrated in Fig. 3. This data is mainly sourced from ShareGPT4V (Chen et al., 2023) and CapRL (Xing et al., 2025), covering diverse modalities and content types, including natural images, AIGC images, tables, and multiple

QA styles such as descriptive and multiple-choice formats.

As our training framework incorporates reinforcement learning to improve generalization, the Open-ended QA data introduces practical challenges. Its free-form outputs are difficult to constrain using structured reward functions, making it only feasible to reliably evaluate fixed output formats and coarse-grained content correctness. To address this issue, we reorganize human-annotated descriptions into **Multiple-choice QA**, as shown in Fig. 4. This conversion transforms open-ended descriptions into deterministic choice-based questions, enabling the model to improve its fine-grained content understanding and analytical ability by selecting the correct options. More details on the construction of Multiple-choice QA are provided in Appendix A.4.1, and representative QA examples are shown in Fig. 3.

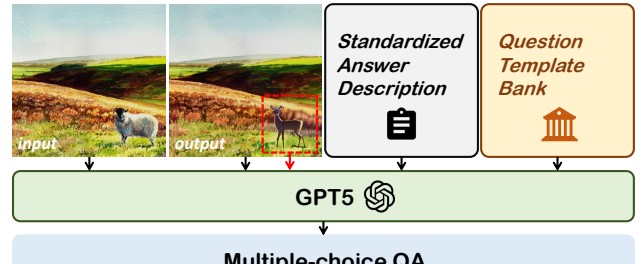

*Figure 4.* Pipeline for Constructing the Multiple-Choice QA Dataset.

However, this strategy alone is far from sufficient. As discussed in Section 3.1 (Data Construction Pipeline), it is difficult to construct and annotate large-scale I2I editing data that contains diverse and realistic errors. In the next Section 4, we will describe in detail how we expand the data scale and enhance model capability through a multi-stage training scheme.

In addition, the existing data scale remains insufficient to effectively improve the weak pixel-level perceptual capability

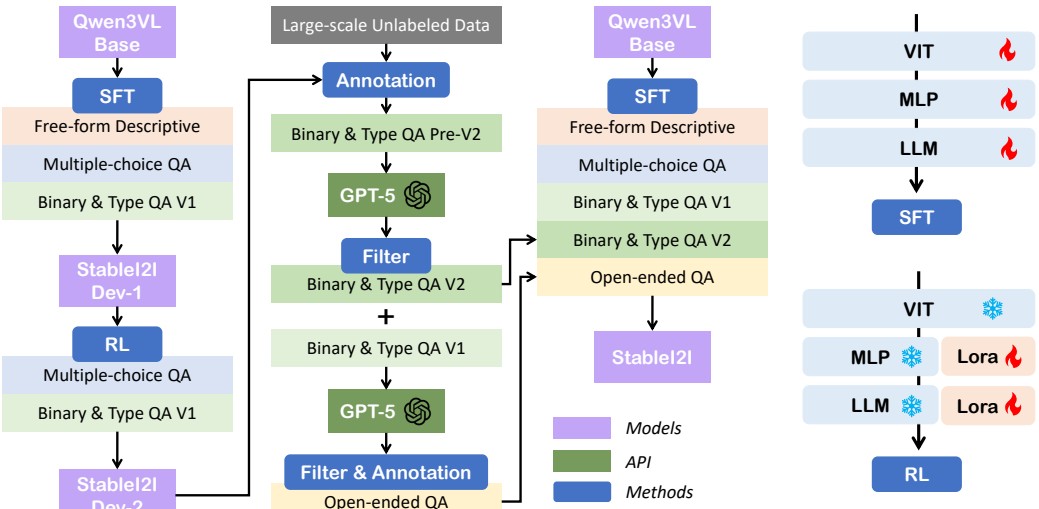

*Figure 5.* The three columns on the left illustrate the training pipeline, including the training strategy at different stages and the corresponding data composition. The single column on the right shows the configuration of trainable model parameters under different training strategies.

of the ViT encoder. We therefore introduce Texture-Aware Enhancement Data to enhance the encoder's perception at the pixel level. Details of its construction pipeline and the data composition of the overall StableI2I-Train dataset are provided in Appendix A.4.1.

## 4. StableI2I

For training, we first perform supervised fine-tuning on Qwen3-VL-8B-Instruct (Team, 2025) using Binary & Type QA, Multiple-choice QA, and Free-form Descriptive data, enabling the model to perform basic task responses while preserving visual perception and comprehension abilities. We then apply reinforcement learning with GRPO on the SFT-trained model to further improve generalization. The rewards are defined separately for Multiple-Choice (MC) tasks corresponding to Multiple-choice QA data, and Binary Answer tasks corresponding to Binary & Type QA data.

For MC tasks, let $G$ and $\hat{G}$ denote the ground-truth and predicted option sets. If $\hat{G} \nsubseteq G$, the reward is zero; otherwise, $R_{\text{MC}} = M \cdot \frac{|\hat{G}|}{|G|}$, where $M$ is the maximum MC reward.

For Binary Answer tasks, each output contains an *answer* and a *problem* field (see Format (1)). We first require the predicted *answer* to exactly match the ground truth; otherwise, the reward is zero. When the ground-truth answer is *Yes*, a reward of 1 is assigned only if both problem sets are empty. When the answer is *No*, both problem sets must be non-empty. Let $P$ and $\hat{P}$ denote the ground-truth and predicted problem type sets. The reward is computed as

$$R_{\text{Binary}} = \max\left(0, \frac{|\hat{P} \cap P|}{|P|} - \alpha\frac{|\hat{P} \setminus P|}{|P|}\right), \quad (1)$$

where $\alpha$ penalizes false positive predictions.

We then use the reinforcement learning–trained model to

annotate large-scale unlabeled data from various I2I tasks. Subsequently, GPT-5 is employed to filter out samples with obvious errors. Next, we combine the original Binary & Type QA data with the newly annotated Binary & Type QA data to construct Open-ended QA data, corresponding to Format (2), where GPT-5 generates the associated *think* and *problem* fields. Finally, we perform full fine-tuning of Qwen3-VL-8B-Instruct using the mixture of all original and newly annotated data to obtain the final model. Through this training paradigm, the model achieves stronger perceptual understanding and improved generalization capability. The overall training pipeline is illustrated in Fig. 5.

## 5. Experiments

**Training Details.** We adopt two types of training paradigms in our experiments: supervised fine-tuning (SFT) and reinforcement learning (RL). For SFT, we use a learning rate of $1 \times 10^{-5}$ with a cosine learning rate scheduler and a warmup ratio of 0.03. The total batch size is set to 128, and the model is trained for 5 epochs over the full training set. For RL, we also use a learning rate of $1 \times 10^{-5}$ together with a cosine scheduler and a warmup ratio of 0.03. A weight decay of 0.01 is applied for regularization. We employ GRPO as the policy optimization algorithm, where 16 candidate responses are generated for each training sample. During training, 8 samples are processed in parallel. The RL experiments are run for at least 5,000 training steps. Unless otherwise specified, both SFT and RL experiments are conducted using 8 NVIDIA H200 GPUs.

### 5.1. Evaluation Results on StableI2I-Bench

Here, we evaluate a range of mainstream open-source and proprietary multimodal large models on our StableI2I-Bench to analyze whether these models can accurately as-

*Table 1.* Quantitative comparison of mainstream models on StableI2I-Bench. Binary Accuracy measures answer correctness, while Strict Accuracy additionally requires correct error types. Best and second-best results are highlighted in **dark blue** and light blue, respectively.

| Models | Binary Accuracy | | | | Strict Accuracy | | | |
|---|---|---|---|---|---|---|---|---|
| | Structure | Semantic | Low-level | Avg. | Structure | Semantic | Low-level | Avg. |
| **Open-Source Models** | | | | | | | | |
| Qwen3VL-8B-Instruct | 36.60 | 55.60 | 81.60 | 57.93 | 13.80 | 31.70 | 63.60 | 36.37 |
| Qwen3VL-32B-Instruct | 53.20 | 73.60 | 87.70 | 71.50 | 34.90 | 48.90 | 56.30 | 46.70 |
| InternVL-3.5-8B | 36.30 | 64.60 | 59.10 | 53.33 | 13.00 | 21.30 | 28.10 | 20.80 |
| InternVL-3.5-38B | 50.10 | 64.90 | 81.70 | 65.57 | 42.30 | 39.40 | 31.60 | 37.77 |
| **Proprietary Models** | | | | | | | | |
| Grok-4.1 | 50.50 | 73.70 | 77.90 | 67.37 | 38.50 | 56.30 | 28.70 | 41.17 |
| Claude-Sonnet-4.5 | 66.20 | 70.10 | 89.70 | 75.33 | 62.40 | 54.40 | 73.30 | 63.37 |
| Claude-Sonnet-4.5-think | 63.80 | 69.40 | 84.70 | 72.63 | 62.50 | 56.20 | 65.20 | 61.30 |
| Gemini-2.5-pro | 66.67 | 79.90 | 90.70 | 79.09 | 56.66 | 58.60 | 37.20 | 50.82 |
| Gemini-3-pro | 71.52 | 83.61 | 91.72 | 82.28 | 62.19 | 63.69 | 75.56 | 67.15 |
| GPT-4o | 59.90 | 79.70 | 94.70 | 78.10 | 46.60 | 60.80 | 71.10 | 59.50 |
| GPT-5 | 65.50 | 83.00 | 93.20 | 80.57 | 54.60 | 60.20 | 51.00 | 55.27 |
| StableI2I | 85.40 | 82.80 | 99.10 | 89.10 | 83.70 | 67.30 | 98.00 | 83.00 |

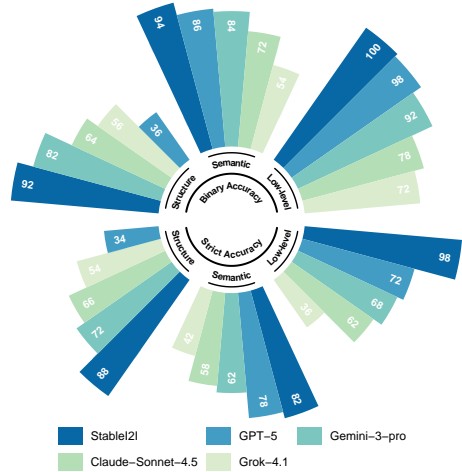

*Figure 6.* Human evaluation of answer accuracy.

sess fidelity in I2I tasks. For open-source models, we select Qwen3VL-8B-Instruct (Team, 2025), Qwen3VL-32B-Instruct (Team, 2025), InternVL-3.5-8B (Wang et al., 2025), and InternVL-3.5-38B (Wang et al., 2025). For proprietary models, we include Grok-4.1 (xAI, 2025), Claude-Sonnet-4.5 (Anthropic, 2025), Claude-Sonnet-4.5-think (Anthropic, 2025), Gemini-2.5-pro (Team et al., 2023), Gemini-3-pro (Team et al., 2023), GPT-4o (Achiam et al., 2023), and GPT-5 (OpenAI, 2025a).

The evaluation results are reported in Tab. 1. And we note an important detail regarding the evaluation setting. The input template used by StableI2I at inference time is not identical to the template provided in StableI2I-Bench for evaluating general-purpose MLLMs. However, both settings use exactly the same image pairs $(I_{in}, I_{out})$ and the same I2I control instruction $x$. This discrepancy arises because StableI2I is a specialized model trained for I2I fidelity assessment, and its training relies on a fixed task template. As a result, compared to general-purpose MLLMs, StableI2I exhibits weaker robustness to template variations when prompts contain additional prior knowledge or more complex instruction structures.

To assess the impact of template priors, we additionally report in Appendix B.1 the results of several mainstream MLLMs evaluated under the simplified StableI2I template. The results show that, after removing the structured priors explicitly provided in the benchmark template, the performance of general-purpose models drops significantly across all three fidelity dimensions and becomes substantially worse than their performance under the original benchmark template.

As shown in Tab. 1, among all mainstream models, Gemini-3-pro achieves the best overall performance. In general, these models perform strongest at the Semantic Level, which is likely because most contemporary models primarily focus on high-level visual information. Moreover, when promi-

nent low-level degradations are present in the output image, models are often able to respond to such issues to some extent. In contrast, most models perform relatively poorly on Structure Level QA. This can be mainly attributed to two factors. First, the task requires pixel-level alignment. Second, in some samples, although the semantic content remains consistent, the global structure has been repainted or altered. After training and fine-tuning, our model outperforms existing state-of-the-art vision models on this task. This result further indicates that there remains substantial room for improvement in current visual models with respect to I2I fidelity assessment.

To verify that our evaluation results align with human priors, we recruited seven volunteers to assess the correctness of the model outputs shown in Fig. 6. Specifically, we randomly selected 50 images generated by Bagel, Nano-Banana and GPT-Image-1 on ImgEdit-Bench, and asked the model to perform evaluations using the response templates defined in StableI2I-Bench. The results indicate that the evaluation outputs produced by StableI2I are largely consistent with human judgments. In addition, we provide supplementary quantitative comparisons with ImgEdit-Judge (Ye et al., 2025) in the Appendix. B.2.

## 5.2. Model Performance on I2I Tasks Assessed by StableI2I

In this section, we use StableI2I to score the performance of mainstream generative models on multiple existing image editing benchmarks as well as on a collection of image restoration tasks. For image editing benchmarks, we adopt ImgEdit-Bench (Ye et al., 2025) and GEdit-Bench (Liu et al., 2025). For low-level restoration tasks, we construct a dataset by sampling data from various scenarios, including denoising, deblurring, deraining, dehazing, and exposure correction. See Appendix A.1 for more details.

The open-source models evaluated include Lumina-

*Table 2.* Quantitative results of mainstream I2I models on image editing and restoration tasks evaluated using StableI2I. The reported values correspond to the accuracy of each evaluation dimension over the entire benchmark. The best-performing results are highlighted in **dark blue**, while the second-best results are highlighted in light blue.

| Datasets | ImgEdit-Bench | | | | GEdit-Bench | | | | Low-level Dataset | | | |
|---|---|---|---|---|---|---|---|---|---|---|---|---|
| | Semantic | Structure | Low-level | Avg. | Semantic | Structure | Low-level | Avg. | Semantic | Structure | Low-level | Avg. |
| **Open-Source Models** | | | | | | | | | | | | |
| Lumina-DiMOO | 0.9366 | 0.2465 | 0.8732 | 0.6854 | 0.7913 | 0.0776 | 0.5677 | 0.4790 | 0.6880 | 0.2740 | 0.4910 | 0.4843 |
| Flux.1-dev | 0.3345 | 0.0123 | 0.9701 | 0.4390 | 0.2368 | 0.0223 | 0.8589 | 0.3727 | 0.2400 | 0.1140 | 0.4590 | 0.2710 |
| OmniGen2 | 0.8803 | 0.6567 | 0.6655 | 0.7342 | 0.8325 | 0.6881 | 0.7294 | 0.7518 | 0.8320 | 0.6600 | 0.5260 | 0.6727 |
| Bagel | **0.9718** | **0.8750** | 0.8046 | **0.8838** | 0.8870 | **0.8003** | 0.7979 | **0.8292** | **0.9520** | **0.9240** | 0.5630 | **0.8130** |
| Qwen-Image-Edit-2509 | 0.9525 | 0.6849 | 0.9718 | 0.8697 | 0.9068 | 0.6271 | 0.9142 | 0.8174 | 0.8480 | 0.6620 | 0.5390 | 0.6830 |
| Qwen-Image-Edit-2511 | 0.9595 | 0.4683 | 0.9349 | 0.7876 | **0.9134** | 0.5899 | 0.8977 | 0.8021 | 0.8720 | 0.6620 | 0.5450 | 0.6930 |
| **Proprietary Models** | | | | | | | | | | | | |
| GPT-Image-1 | 0.8390 | 0.1342 | **0.9839** | 0.6524 | 0.6160 | 0.0693 | **0.9182** | 0.5347 | 0.7333 | 0.0283 | 0.4717 | 0.4111 |
| Nano-Banana | 0.9665 | 0.6772 | 0.9506 | 0.8648 | 0.8803 | 0.5070 | 0.8908 | 0.7594 | 0.8878 | 0.7455 | 0.5591 | 0.7308 |

*Figure 7.* Qualitative results of mainstream I2I models on image editing and restoration tasks evaluated using StableI2I. From top to bottom, the three groups of examples are drawn from ImgEdit-Bench, GEdit-Bench, and the Low-level Dataset, respectively. Qwen-Image-Edit refers to the Qwen-Image-Edit-2511 model release. For each evaluation dimension in StableI2I, an output of "Yes" indicates no detected error, whereas "No" denotes the presence of an error, accompanied by a brief problem type describing the corresponding inconsistency. Please zoom in for better visualization of fine-grained details.

DiMOO (Xin et al., 2025), Flux.1-dev (Labs et al., 2025), OmniGen2 (Wu et al., 2025b), Bagel (Deng et al., 2025), Qwen-Image-Edit-2509 (Wu et al., 2025a), and Qwen-Image-Edit-2511 (Wu et al., 2025a), while the proprietary models include GPT-Image-1 (Wu et al., 2025a) and Nano-Banana (Google, 2025).

The scoring protocol measures the proportion of samples for which StableI2I answers "Yes" (i.e., no fidelity issues detected) in each evaluation dimension, relative to the total number of samples. Detailed results are reported in Tab. 2.

From the above table, we find that Bagel and the Qwen series demonstrate superior performance in terms of fidelity. In general, most models achieve acceptable results at the semantic level, suggesting that they can largely preserve high-level semantic information. In contrast, their performance at the structure level is markedly worse, indicating a limited ability to maintain fine-grained structural consistency. Notably, Flux.1-dev and GPT-Image-1 both suffer from severe content repainting in real-world evaluations, where the original structural layout is largely discarded and regenerated. This behavior leads to extremely low Structure

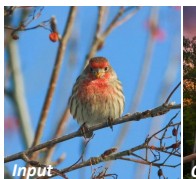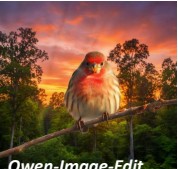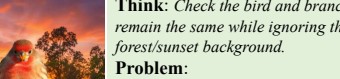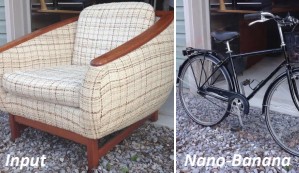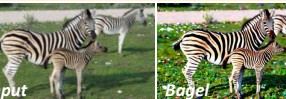

**Think**: *Check the bird and branch remain the same while ignoring the new forest/sunset background.*
**Problem**:
"replace": Bird and perching branch look different (plumage/pose and branch thickness/shape changed).

**Prompt:** Change the background from a clear blue sky with bare branches to a forest sunset sky.

**Think**: *Interpretation: Because the task prompt is a semantic edit instruction, low-level degradations are out-of-scope unless explicitly stated. Thus, the preservation requirement applies to low-level characteristics in non-target regions: they should not worsen or drift systematically.*
**Problem**:
"color cast": Cool color cast (medium).

**Prompt:** Replace the chair in the image with a bicycle.

**Think**: *Check that the person, shelf/table, teapot, and background remain unchanged, ignoring the plant removal and enlarged frame.*
**Problem**:
"remove": The shelf/table with teapot and other items disappeared.

**Prompt:** Remove the plant from the shelf, and resize the picture frame to be larger.

**Think**: *Check that zebras and scene layout stay the same while ignoring color/exposure changes.*
**Problem**:
"add": Colorful flower petals and leaves were added across the grass, turning it from plain to multicolored.

**Prompt:** Restore the image.

*Figure 8.* This figure presents representative failure cases of different models on ImgEdit-Bench, together with a detailed analysis of the observed errors. Zooming in is recommended for better visualization of fine-grained details.

Level scores for both models.

Fig. 7 provides a qualitative overview of StableI2I's evaluations across image editing and restoration tasks. We observe that all three types of information drift can occur in real-world scenarios, with unintended content repainting being the most critical issue for current generative models. Fig. 8 further illustrates StableI2I's detailed error descriptions. Since structural errors correspond to global changes with limited categories, we focus on a finer-grained analysis of Semantic Level and Low-level Appearance, along with their associated error types.

### 5.3. Ablation Study

*Table 3.* The following shows a quantitative ablation study on the use of Multiple-Choice QA. The reported values are the accuracy of samples where both the answer and the problem type in Binary & Type QA are predicted correctly.

| SFT | | RL | | Binary & Type QA | | | |
|---|---|---|---|---|---|---|---|
| w/o | w/ | w/o | w/ | Structure | Semantic | Low-level | Avg. |
| | | ✓ | | 53.50 | 21.40 | 24.60 | 33.17 |
| | ✓ | ✓ | | 63.90 | 65.10 | 65.80 | 64.95 |
| ✓ | | ✓ | | 79.50 | 67.80 | 94.10 | 80.47 |
| ✓ | | | ✓ | 81.10 | 67.10 | 94.00 | 80.73 |
| | ✓ | | ✓ | 81.80 | 68.90 | 95.00 | 81.90 |

We demonstrate that incorporating **Multiple-Choice QA** can effectively enhance a model's capability on fundamental tasks. Tab. 3 illustrates the impact of introducing Multiple-Choice QA on the performance of **Binary & Type QA**.

From top to bottom, the rows correspond to: (1) the capability of the base model; (2) the performance obtained by applying RL on the base model *without* using multiple-choice questions; (3) the performance of first performing SFT and then RL, while *not* using multiple-choice questions in either stage; (4) the performance of first performing SFT and then RL, where multiple-choice questions are introduced only in the RL stage; and finally, (5) the performance where multiple-choice questions are used in both stages.

We observe that Multiple-Choice QA effectively transforms open-ended content descriptions into fixed-choice questions,

and that training with such questions substantially improves the model's perceptual capability. All experiments are conducted under identical parameter settings and with the same number of training steps.

*Table 4.* The following shows the accuracy of models at different stages of multi-stage training on Multiple-Choice QA (MCQ) and Binary & Type QA. For Binary & Type QA, the reported values are the accuracy of samples where both the answer and the problem type are predicted correctly.

| Models | MCQ | Binary & Type QA | | | Total |
|---|---|---|---|---|---|
| | Avg. | Structure | Semantic | Low-level | |
| Base | 41.90 | 53.50 | 21.40 | 24.60 | 37.53 |
| StableI2I-Dev.1 | 92.37 | 83.70 | 59.00 | 93.70 | 85.58 |
| StableI2I-Dev.2 | 91.17 | 80.30 | 67.70 | 96.60 | 86.35 |
| StableI2I | 91.70 | 83.70 | 67.30 | 98.00 | 87.35 |

Tab. 4 shows the effect of our multi-stage training and data augmentation strategy on model performance, and the overall training pipeline is illustrated in Fig. 5. StableI2I-Dev.1 denotes the first-stage SFT model. Owing to the limited amount of editing data, it performs poorly on the **Semantic** dimension. After RL training, StableI2I-Dev.2 significantly improves the accuracy on this dimension; however, the reduced diversity of RL data leads to performance degradation on several other categories. By combining the augmented data used in StableI2I-Dev.2 with the first-stage data and re-training the base model via SFT, we obtain the final model StableI2I. Compared with StableI2I-Dev.1, StableI2I achieves substantially better overall performance, while largely retaining the gains of StableI2I-Dev.2. These results confirm the effectiveness of our data augmentation strategy in improving overall model performance.

## 6. Conclusion

StableI2I is the first framework to systematically evaluate fidelity in image-to-image (I2I) tasks from both semantic and pixel-level perspectives. It enables reliable assessment of whether generative models preserve critical visual information and provides StableI2I-Bench as a precise benchmark

for evaluating MLLMs under multi-image consistency constraints. By requiring consistency across multiple images at both semantic and pixel levels, this benchmark poses a substantial challenge to existing MLLMs and serves as an effective tool for measuring perceptual ability beyond single-image understanding. We believe that the introduction of StableI2I can substantially improve the generation quality of I2I models, leading to more faithful, realistic, and perceptually consistent outputs.

**Limitations.** For detailed subjective visualizations and analysis of specific failure cases, please refer to Appendix B.4.

## Acknowledgments

This work was supported in part by the Shenzhen Science and Technology Program (JCYJ20241202125904007, SYSPG20241211173440004), the Guangdong Provincial Key Laboratory of Ultra High Definition Immersive Media Technology (2024B1212010006), the Outstanding Talents Training Fund in Shenzhen, and Shanghai Artificial Intelligence Laboratory.

## Impact Statement

This paper introduces StableI2I, a unified evaluation framework designed to enhance the reliability of image-to-image (I2I) transitions by detecting unintended semantic and structural drift. By establishing a principled methodology for measuring content fidelity across semantic, structural, and low-level appearance dimensions without requiring reference images, our work provides a critical diagnostic tool for high-stakes applications such as medical imaging and remote sensing, where information consistency is paramount. While StableI2I facilitates the development of more trustworthy generative systems, we acknowledge that automated fidelity assessments must be continually audited for potential biases in "consistency" definitions to ensure the framework remains inclusive of diverse visual domains and cultural contexts.

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

# A. Dataset

This section mainly provides supplementary details on the data sources, the models used for data construction, the human annotation workflow, and the prompts adopted in the data generation process.

### A.1. Data Construction and Statistics

Our data pipeline primarily leverages images from ImageNet (Deng et al., 2009), Unsplash (Unsplash, 2025), DIV2K (Agustsson & Timofte, 2017), Alchemist (Startsev et al., 2025) and ArtiMuse (Cao et al., 2025b). The exact number of images drawn from each source is reported in Tab. 6. After collection, we first constrain the maximum side length of each image to be no greater than 1344 pixels; images exceeding this limit are resized with preserved aspect ratio.

For image restoration tasks, we apply the ESRGAN (Wang et al., 2018) degradation pipeline to synthesize degraded inputs. We then perform text-guided image restoration using three models: SeeSR (Wu et al., 2024b), SUPIR (Yu et al., 2024), and OSEDiff (Wu et al., 2024a). To further increase the diversity of generated samples, a subset of the restored images is subjected to a second-stage enhancement, mainly using SwinIR (Liang et al., 2021) and ESRGAN (Wang et al., 2018). For restoration results obtained under real-world settings, we use PromptIR (Potlapalli et al., 2023), OSEDiff (Wu et al., 2024a), Qwen-Image-Edit-2509 (Wu et al., 2025a), and Bagel (Deng et al., 2025). Specifically, we constructed our Low-level Dataset by randomly sampling 500 images from several public datasets and applying the ESRGAN degradation pipeline, as detailed in Tab. 5.

*Table 5.* Constuction of the Low-level Dataset.

| Type | Source Dataset | # Sample |
|------|----------------|----------|
| Low-Light Enhancement | LOL (Wei et al., 2018) | 409 |
| Image Dehazing | Nature20 (Li et al., 2019b) | 190 |
| Image Dehazing | O-Haze (Ancuti et al., 2018) | 40 |
| Deraining | Rain800 (Yasarla et al., 2020) | 661 |
| Raindrop Removal | RainDrop (Qian et al., 2018) | 861 |
| Underwater Image Enhancement | UIEB (Li et al., 2019a) | 700 |
| Image Super-Resolution | DIV2K (Agustsson & Timofte, 2017) | 800 |
| **Total** | | **3661** |

For image editing tasks, the data are generated using Qwen-Image-Edit-2509 (Wu et al., 2025a), OmniGen (Xiao et al., 2025), SD3 (Esser et al.), and GPT-Image-1 (OpenAI, 2025b). Our editing instructions are primarily constructed by prompting GPT-5 to integrate the visual information in the image and to formulate edits based on three major categories—add, replace, and remove—resulting in a concise editing instruction. The number of samples constructed in the first stage is also reported in Tab. 6.

*Table 6.* Data usage and annotation statistics. *Total* denotes the number of intact images after download and removal of corrupted files. *Image Restoration* and *Image Editing* indicate the numbers of successfully synthesized samples produced by their respective pipelines. **In Image Restoration**, *GPT-5* refers to samples initially annotated using GPT-5, selected from the synthesized data and successfully labeled. *Human* denotes the subset randomly sampled from the GPT-5–annotated data (15,000 samples) and subsequently cleaned and verified by human annotators. **In Image Editing**, *V1-Human* denotes the number of samples annotated by human annotators in the first stage, and *V2-Enhance* denotes the number of samples annotated using StableI2I-Dev.2 after multi-stage training.

| Source | Total | Image Restoration | GPT-5 | Human | Image Editing | V1-Human | V2-Enhance |
|--------|-------|-------------------|-------|-------|---------------|----------|------------|
| ArtiMuse | 10002 | 4587 | | | 9933 | – | 9933 |
| ImageNet | 136590 | 89490 | | | – | – | – |
| DIV2K | 800 | 800 | 70640 | 6722 | 800 | 800 | – |
| Unsplash | 24996 | 19264 | | | 24996 | – | 24996 |
| Alchemist | 4039 | 4039 | | | 4039 | 4039 | – |

## A.2. Human Annotation and Annotator Training

We first issued a public tender for the annotation task, and three professional annotation companies submitted bids. For evaluation, we selected 100 samples for each task and provided detailed annotation guidelines, asking all three companies to conduct a pilot annotation. The pilot annotation accuracies achieved by the three companies were 0.852, 0.725, and 0.700, respectively. We selected the company with the highest accuracy to carry out the full-scale annotation.

The final dataset consists of two parts: image restoration data (15,000 samples randomly drawn from GPT-5 coarse annotations) and image editing data (4,839 samples). Each task was annotated by a team of 10 annotators over seven consecutive working days, using a cross-annotation protocol that included one round of annotation followed by a review phase. Prior to annotation, all annotators received video-based training. During the annotation process, questions were addressed in real time through a shared document.

For the image restoration data, annotators were only required to judge whether GPT-5's answer was correct and to label the level of degradation using a three-point scale. For the image editing data, annotators were required to assign an error type, optionally use bounding boxes to localize the problematic regions when necessary, and finally label the level of degradation using the same three-point scale.

In total, we obtained 6,722 human-annotated image restoration samples and 4,839 image editing samples. At acceptance, the overall annotation pass accuracy reached 94%.

## A.3. Detailed Description of StableI2I-Bench

All data in StableI2I-Bench are based on human-annotated samples, and the dataset has no overlap with the data used in Stable-Train.

We first present the StableI2I-Bench evaluation templates, followed by representative examples from the benchmark.

### A.3.1. STABLEI2I-BENCH EVALUATION TEMPLATES

We adopt a **fixed prompt template** for all evaluation samples. Each prompt consists of four parts: (1) two input images, (2) the corresponding I2I instruction (prompt), (3) a description of the prerequisite knowledge, and (4) a specification of the required output format. The benchmark covers three evaluation dimensions: **Semantic Level**, **Structure Level**, and **Low-level Appearance**. Each dimension contains 1000 image pairs, and within each dimension, the proportion of samples labeled as "no issue" does not exceed 50%. Below we provide detailed descriptions of the prompt design for the three evaluation dimensions.

---

**Semantic Level**

**The first image** `<image>`: Before processing.
**The second image** `<image>`: After processing.

**The task prompt is:** `<TASK_PROMPT>`

**Task Guidelines:**
```
Please evaluate this image-to-image (I2I) transition from a semantic content fidelity
perspective.
Compare the output image strictly against the input image, conditioned on the given
task prompt.  Your goal is to determine whether any regions that should remain
unchanged have undergone unintended semantic changes.
Specifically, check whether the output image introduces any of the following semantic
inconsistencies relative to the input image:
ADD: new objects, parts, text, symbols, or meaningful elements appear that are not
implied by the task prompt.
REMOVE: existing objects, parts, text, or meaningful elements in the input image are
missing in the output image.
REPLACE: an existing object, part, or attribute is substituted with a different
semantic entity (e.g., a dog becomes a cat), or a meaningful attribute changes (e.g.,
``red'' becomes ``blue'') when such change is not implied by the task prompt.
IMPORTANT GUIDELINES: 1) Focus on semantic content only.  Ignore purely low-level
appearance differences (e.g., mild noise or compression artifacts) unless they
```

---

cause an actual semantic change (e.g., text becomes unreadable). 2) Legitimate
global side effects that are a physically plausible consequence of the intended edit
(e.g., shadows, reflections, or minor lighting changes) should not be counted as
semantic errors. 3) If the task prompt is NULL (no specified edit or restoration
intent), then the expected behavior is identity mapping: the two images should be
completely identical in semantic content. Any semantic difference should be marked as
inconsistent. 4) Use "No" whenever you detect any potential semantic inconsistency in
regions that should have been preserved.

**Output Format:**
Return your decision in a single line of valid JSON with the format: {"answer":
"Yes", "problem": "NULL"} if the images are semantically consistent, otherwise
{"answer": "No", "problem": ["add", "replace", "remove"]}.

---

**Structure Level**

**The first image** <image>: Before processing.
**The second image** <image>: After processing.

**The task prompt is:** <TASK_PROMPT>

**Task Guidelines:**
Please evaluate this image-to-image (I2I) transition from a structural and texture
consistency perspective.
Compare the output image strictly against the input image, conditioned on the given
task prompt. Your goal is to determine whether any regions that are expected to
remain unchanged have undergone unintended structural or texture-level changes.
Specifically, check for the following types of inconsistencies:
Misalignment: global or local geometric distortions, spatial shifts, shape warping,
incorrect object boundaries, layout drift, or broken structural coherence relative to
the input image.
Repainting: unintended re-rendering of textures, materials, fine surface details,
or local appearance patterns in regions that should have been preserved (e.g., skin,
background, clothing, walls, ground, hair texture outside the edited region).
Important guidelines: 1) Only judge regions that are NOT explicitly targeted by the
task prompt. Any change that is a necessary and physically plausible consequence
of the intended edit (e.g., lighting, shading, subtle color adaptation) should
NOT be counted as an error. 2) Focus on structure and texture consistency only.
Ignore purely semantic category changes unless they manifest as clear repainting or
structural deformation. 3) If the task prompt is NULL (no specified edit/restoration
intent), then the expected behavior is identity mapping: the two images should be
completely identical in structure and texture. Any deviation should be marked as
inconsistent. 4) If both misalignment and repainting are observed, list both. 5)
When uncertain, choose ``No''(i.e., favor sensitivity over specificity).

**Output Format:**
Return your decision in a single line of valid JSON with the format: {"answer":
"Yes", "problem": "NULL"} if the images are consistent, otherwise {"answer": "No",
"problem": ["misalignment", "repainting"]}, where the "problem" field should reflect
the dominant issue(s) observed.

---

**Low-level Appearance**

**The first image** <image>: Before processing.
**The second image** <image>: After processing.

**The task prompt is:** <TASK_PROMPT>

**Task Guidelines:**
Please evaluate this image-to-image (I2I) transition from a low-level visual fidelity
perspective.
Compare the output image strictly against the input image and determine whether
the processing has introduced any unintended low-level visual degradation or
distributional shift.

```
    Specifically, check for the presence of any of the following issues in the output
    image relative to the input image:
    Blur:  loss of sharpness or edge detail,
    Noise:  newly introduced random pixel-level noise or grain,
    Color cast:  unintended global or local color shifts,
    Exposure degradation:  over-exposure, under-exposure, or brightness/contrast
    distortion,
    Artifact:  compression artifacts, ringing, blocking, haloing, or other synthetic
    patterns.
    This task focuses only on unintended low-level changes.  Do NOT consider high-level
    semantic differences or structural changes.
    If the processing described in the task prompt is explicitly intended to produce any
    of the above effects (e.g., denoising, deblurring, color correction, artifact removal,
    exposure adjustment), then this case should be ignored.
    If no specific task type is given (i.e., the task prompt is NULL), simply judge
    whether the two images are pixel-wise and perceptually identical, up to negligible
    numerical or compression differences.
```
**Output Format:**
```
Return your decision in a single line of valid JSON with the format:  In the ``ignored"
case, output:  {"answer":  "NULL", "problem":  "NULL"} If the output image is
consistent with the input image at the low-level appearance:  {"answer":  "Yes",
"problem":  "NULL"} Otherwise, if any unintended low-level degradation or shift is
detected: {"answer":  "No", "problem":  ["noise", "blur", "color cast", "exposure
degradation", "artifact"]}
```

### A.3.2. STABLEI2I-BENCH EXAMPLES

Below, we randomly select three benchmark entries for illustration, corresponding to examples at the Structure level, Semantic level, and Low-level Appearance level, respectively.

---

**StableI2I-Bench: Structure Level I2I Evaluation Example**

**ID:** test_30

**Images:**
```
Input image:  /XXX/alchemist/image/9900644e50f1a52f5d5bb172e5bda971.jpg
Output image:  /XXX/alchemist/image_edit/9900644e50f1a52f5d5bb172e5bda971.jpg
```

**Human Prompt:**
```
The first image <image>:  Before processing.
The second image <image>:  After processing.
The task prompt is:  Replace the gray jacket with a light pastel-colored cardigan to
complement the spring setting.
Please evaluate this image-to-image (I2I) transition from a structural
and texture consistency perspective.

Compare the output image strictly against the input image, conditioned
on the given task prompt.  Your goal is to determine whether any regions
that are expected to remain unchanged have undergone unintended
structural or texture-level changes.

Specifically, check for the following types of inconsistencies:

 - misalignment:  global or local geometric distortions, spatial shifts,
 shape warping, incorrect object boundaries, layout drift, or broken
 structural coherence relative to the input image.

 - repainting:  unintended re-rendering of textures, materials, fine
 surface details, or local appearance patterns in regions that should
 have been preserved (e.g., skin, background, clothing, walls, ground,
 hair texture outside the edited region).
```

```
Important guidelines:
1) Only judge regions that are NOT explicitly targeted by the task
prompt.  Any change that is a necessary and physically plausible
consequence of the intended edit (e.g., lighting, shading, subtle
color adaptation) should NOT be counted as an error.
2) Focus on structure and texture consistency only.  Ignore purely
semantic category changes unless they manifest as clear repainting
or structural deformation.
3) If the task prompt is NULL (no specified edit/restoration intent),
then the expected behavior is identity mapping:  the two images should
be completely identical in structure and texture.  Any deviation
should be marked as inconsistent.
4) If both misalignment and repainting are observed, list both.
5) When uncertain, choose ``No'' (i.e., favor sensitivity over specificity).

Return your decision in a single line of valid JSON with the format:
{"answer":  "Yes", "problem":  "NULL"} if the images are consistent,
otherwise {"answer":  "No", "problem":  ["misalignment", "repainting"]},
where the "problem" field should reflect the dominant issue(s) observed.
```

**Model Output (GT):**
```
{"answer":  "No", "problem":  ["repainting"]}
```

---

**StableI2I-Bench: Semantic Level I2I Evaluation Example**

**ID:** test_1589

**Images:**
```
Input image:  /XXX/imagenet/lq_input/n02396427_11580.jpg
Output image:  /XXX/imagenet/imagenet100_new_x8_out2_select_15_combine/n02396427_11580.jpg
```

**Human Prompt:**
```
The first image <image>:  Before processing.
The second image <image>:  After processing.
The task prompt is:  Image restoration.
Please evaluate this image-to-image (I2I) transition from a semantic
content fidelity perspective.

Compare the output image strictly against the input image, conditioned
on the given task prompt.  Your goal is to determine whether any regions
that should remain unchanged have undergone unintended semantic changes.

Specifically, check whether the output image introduces any of the
following semantic inconsistencies relative to the input image:
- add:  new objects, parts, text, symbols, or meaningful elements appear
that are not implied by the task prompt.
- remove:  existing objects, parts, text, or meaningful elements in the
input image are missing in the output image.
- replace:  an existing object/part/attribute is substituted with a
different semantic entity (e.g., a dog becomes a cat), or a
meaningful attribute changes (e.g., ``red'' becomes ``blue'')
when such change is not implied by the task prompt.

Important guidelines:
1) Focus on semantic content only.  Ignore purely low-level appearance
differences (e.g., mild noise, compression artifacts) unless they
cause an actual semantic change (e.g., text becomes unreadable).
2) Legitimate global side effects that are a physically plausible
consequence of the intended edit (e.g., shadows, reflections, minor
lighting changes) should NOT be counted as semantic errors.
3) If the task prompt is NULL (no specified edit/restoration intent),
then the expected behavior is identity mapping:  the two images should
```

```
be completely identical in semantic content.  Any semantic difference
should be marked as inconsistent.
4) Use ``No'' whenever you detect any potential semantic inconsistency in
regions that should have been preserved.

Return your decision in a single line of valid JSON with the format:
{"answer":  "Yes", "problem":  "NULL"} if the images are semantically consistent,
otherwise {"answer":  "No", "problem":  ["add", "replace", "remove"]}.
```

**Model Output (GT):**
```
{"answer":  "No", "problem":  ["replace"]}
```

---

**StableI2I-Bench: Low-level Appearance I2I Evaluation Example**

**ID:** test_2922

**Images:**
```
Input image:  /xxx/Low-level/ArtiMuse/pre/0_158.jpg
Output image:  /xxx/Low-level/ArtiMuse/post/0_158.jpg
```

**Human Prompt:**
```
The first image <image>:  Before processing.
The second image <image>:  After processing.
The task prompt is:  Modify the outfit to have a matte finish instead of a glossy one
for a more subdued appearance..
Please evaluate this image-to-image (I2I) transition from a low-level
visual fidelity perspective.

Compare the output image strictly against the input image and determine
whether the processing has introduced any unintended low-level visual
degradation or distributional shift.

Specifically, check for the presence of any of the following issues in
the output image relative to the input image:
- blur:  loss of sharpness or edge detail,
- noise:  newly introduced random pixel-level noise or grain,
- color cast:  unintended global or local color shifts,
- exposure degradation:  over-exposure, under-exposure, or brightness/contrast
distortion,
- artifact:  compression artifacts, ringing, blocking, haloing, or other synthetic
patterns.

This task focuses only on unintended low-level changes.  Do NOT consider
high-level semantic differences or structural changes.

If the processing described in the task prompt is explicitly intended to
produce any of the above effects (e.g., denoising, deblurring, color
correction, artifact removal, exposure adjustment), then this case
should be ignored.

If no specific task type is given (i.e., the task prompt is NULL),
simply judge whether the two images are pixel-wise and perceptually
identical, up to negligible numerical or compression differences.

Return your decision in a single line of valid JSON with the format:

In the ``ignored'' case, output:
{"answer":  "NULL", "problem":  "NULL"}

If the output image is consistent with the input image at the low-level
appearance:
{"answer":  "Yes", "problem":  "NULL"}
```

```
Otherwise, if any unintended low-level degradation or shift is detected:
{"answer":  "No", "problem":  ["noise", "blur", "color cast", "exposure degradation",
"artifact"]}
```

**Model Output (GT):**
```
{"answer":  "No", "problem":  ["blur"]}
```

## A.4. Detailed Description of StableI2I-Train

### A.4.1. SUPPLEMENTARY DETAILS ON DATA SYNTHESIS AND SCALE OF STABLEI2I-TRAIN

Here, we first provide supplementary explanations regarding the **Multiple-Choice QA** component. Since the earlier annotation process already identifies whether an error occurs and specifies the corresponding error type, we have sufficient information to construct multiple-choice questions.

The multiple-choice questions are mainly divided into two categories. The first category is **"Type"**, which focuses on identifying the type of error. The second category is **"Subtype"**, which focuses on identifying the object or region where the error occurs.

The first category of questions is relatively straightforward to construct. We can generate multiple-choice questions by enumerating all error types associated with the given category, together with an additional option indicating that none of the above answers is correct. Based on the error types involved in each sample, we then construct a multiple-selection question with a variable number of correct choices.

In contrast, the second category requires the model to understand which objects or regions are present in the image. Based on this understanding, we construct multiple-choice questions that include both objects that are incorrectly affected and objects that appear in the image but remain unaffected. This process requires the model to perform image understanding and question generation jointly.

The following two JSON examples illustrate the first and second categories of multiple-choice questions, respectively.

---

**Multiple-Choice QA Example (Type)**

**ID:** test_3674

**Images:**
```
Input image:   /XXX/DIV2K/DIV2K_train_HR/0781.png
Output image:  /XXX/DIV2K/DIV2K_train_HR_edit/0781.png
```

**Human Prompt:**
```
The first image <image>:  Before processing.
The second image <image>:  After processing.

Pick one answer.  Given the request to image edit and Edit instruction is Add a pair
of aviator sunglasses to the giraffe's face, which types of unintended semantic issues
occurred in unchanged regions?

A. misalignment
B. No error observed in areas expected to remain unchanged
C. repainting

Return exactly ONE line of JSON with the format:
{"answer":["A","B",...]}
Examples:
Single-choice -> {"answer":["D"]}
Multi-choice -> {"answer":["A","C"]}

Rules:
- "answer" MUST be a list even for single choice.
- Letters MUST be uppercase and chosen ONLY from the options shown.
- Output JSON only.  No extra text.
```

---

**Model Output (GT):**
```
{"answer": ["C"]}
```

---

**Multiple-Choice QA Example (Subtype)**

**ID:** `test_2053`

**Images:**
```
Input image:  /XXX/imagenet/lq_input/n02099849_1606.jpg
Output image:  /XXX/imagenet/imagenet100_new_x8_out2_select_15_combine/n02099849_1606.jpg
```

**Human Prompt:**
```
The first image <image>:  Before processing.
The second image <image>:  After processing.

Pick one answer.  Given the request to image restoration, what element was replaced
even though it should have stayed unchanged in areas expected to remain unchanged?

A. Replacement involving dog → deer
B. Disappearance of background trees
C. Removal of grass
D. Shrubs missing

Return exactly ONE line of JSON with the format:
{"answer":["A","B",...]}
Examples:
Single-choice -> {"answer":["D"]}
Multi-choice -> {"answer":["A","C"]}

Rules:
- "answer" MUST be a list even for single choice.
- Letters MUST be uppercase and chosen ONLY from the options shown.
- Output JSON only.  No extra text.
```

**Model Output (GT):**
```
{"answer": ["A"]}
```

---

During training, we observed that existing multimodal large models exhibit weak pixel-level alignment capability. Therefore, in addition to the standard image restoration and image editing data, we introduced two auxiliary data types to enhance the perceptual capacity of the model encoder: **Texture-Aware Enhancement Data** and **Degraded Image Data**, as illustrated in Fig. 9.

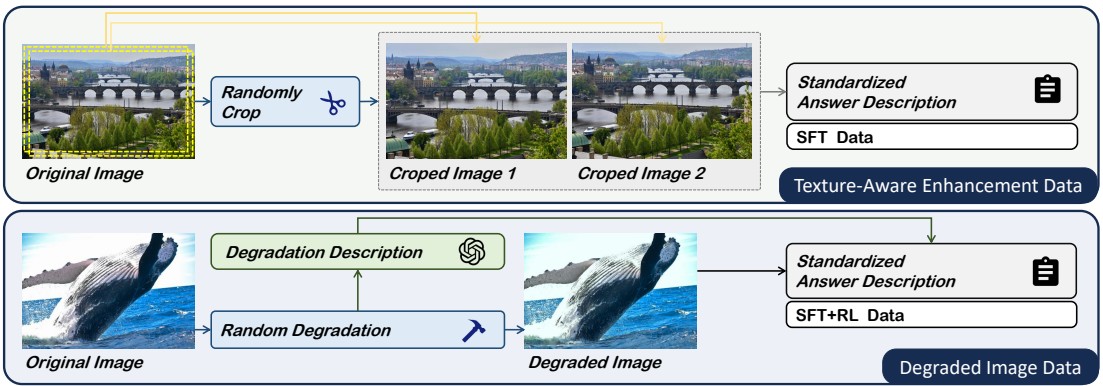

*Figure 9.* An overview of the data construction pipelines for Texture-Aware Enhancement Data and Degraded Image Data.

Texture-Aware Enhancement Data are constructed by randomly cropping a natural image into two different images with a cropping ratio of 95–98%. After this processing, the two images remain nearly identical in global semantic content, but the model is forced to attend to subtle pixel-level structural differences between them. This encourages the model to

learn fine-grained pixel alignment and correspondence. Degraded Image Data are designed to assist the model in learning degradation phenomena that arise in image-to-image (I2I) tasks. This subset covers multiple types of low-level appearance shifts, including blur, noise, compression artifacts, and color distortions.

*Table 7.* The training data volume for the **Binary & Type QA** category is summarized as follows. Real denotes samples generated by the data synthesis pipeline with human annotations (Fig. 2), while Synthetic refers to samples synthesized using the pipeline in Fig. 9. The latter subset is mainly introduced to enhance the model's pixel-level perceptual capability. "–" denotes not applicable.

| Dimension | Source | Image Identity | Image Restoration | Image Editing |
|---|---|---|---|---|
| Structure Level | Synthetic | 207,866 | 415,732 | – |
| | Real | – | 6,722 | 38,904 |
| Semantic Level | Real | – | 70,640 | 54,447 |
| Low-level Appearance | Synthetic | 118,808 | 7,905 | 38,238 |

*Table 8.* The training data volume for the **Multiple-choice QA** category is reported as follows. **Type** refers to answering the type of error that occurs, while **Subtype** refers to answering the specific object or content involved in the error. All questions are formulated as multiple-answer multiple-choice questions. "–" denotes not applicable.

| Dimension | QA Level | Image Identity | Image Restoration | Image Editing |
|---|---|---|---|---|
| Structure Level | Type | – | 5,722 | 3,975 |
| Semantic Level | Type | – | 8,406 | 4,851 |
| | Subtype | – | 7,992 | 1,451 |
| Low-level Appearance | Type | – | 5,186 | 2,995 |
| | Subtype | 2,000 | 2,575 | – |

*Table 9.* The training data volume for the **Open-ended QA** category is summarized as follows. Synthetic denotes answer pairs synthesized from the annotated errors in the data construction pipeline and their corresponding fine-grained error descriptions, while GPT-5 refers to samples whose *think* rationales and answers to fine-grained error questions are generated by GPT-5. "–" denotes not applicable.

| Dimension | Source | Image Restoration | Image Editing |
|---|---|---|---|
| Semantic Level | Synthetic | 70,640 | 4,851 |
| | GPT-5 | 8,406 | 34,929 |
| Low-level Appearance | Synthetic | 8,599 | 34,929 |

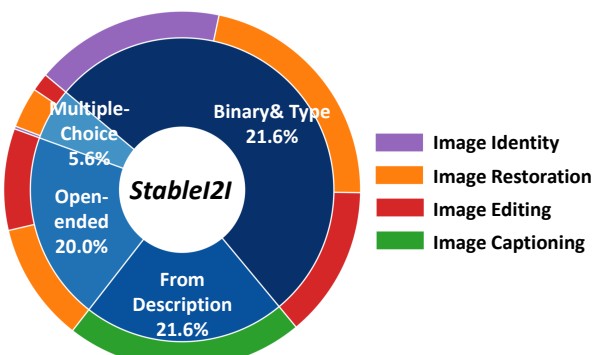

*Figure 10.* The proportions of the four types of training data used for SFT. The overall taxonomy of image-to-image (I2I) tasks is divided into three categories: Image Editing, Image Restoration, and Image Identity, where Image Identity refers to directly comparing two images without applying any transformation.

Building upon these data types, we further introduce an additional task, termed Image Identity, which requires the model to determine whether two identical images exhibit any differences. This task is designed to strengthen the model's ability to perceive fine-grained pixel-level correspondences across multiple images. The images for this task are constructed from the COCO (Lin et al., 2014) dataset following the same pipelines used for Texture-Aware Enhancement Data and Degraded Image Data.

The final data composition ratios at the SFT stage are summarized as follows. The number of Free-form Descriptive samples is 174,866. The statistics of Binary & Type QA are reported in Tab. 7, those of Open-ended QA are reported in Tab. 9, and those of Multiple-choice QA are reported in Tab. 8. The overall proportion of the training data is illustrated in Fig. 10.

### A.4.2. INPUT PROMPT TEMPLATE OF STABLEI2I-TRAIN

During training, we adopt a fixed prompt template to better adapt the model to the target tasks. **Binary & Type QA (Format 1)** covers three evaluation dimensions: Semantic Level, Structure Level, and Low-level Appearance. **Open-ended QA (Format 2)** covers two dimensions: Semantic Level and Low-level Appearance, since the Structure Level corresponds to global changes and does not admit more fine-grained textual descriptions.

We next describe the prompt designs for the three dimensions in Format 1, followed by the two prompt designs for the dimensions in Format 2.

---

**Format 1: Structure Level**

**The first image** `<image>`: Before processing.
**The second image** `<image>`: After processing.

**The task prompt is:** `<TASK_PROMPT>`

**Task Guidelines:**
```
Please determine whether the texture-consistent regions that should remain unchanged
between the pre- and post-process images are indeed consistent (e.g., unchanged areas
remain identical, or for restoration tasks, whether the overall texture and color are
consistent).  If no specific task type is given, simply judge whether the two images
are identical.  Major inconsistencies generally fall into two categories:  structural
misalignment, texture repainting.
```
**Output Format:**
```
Return your decision in a single line of valid JSON with the format:  {"answer":
"Yes", "problem":  "NULL"} if the images are consistent, otherwise {"answer":  "No",
"problem":  ["misalignment", "repainting"]}, where the "problem" field should reflect
the most dominant issue observed between the two images.
```

---

**Format 1: Semantic Level**

**The first image** `<image>`: Before processing.
**The second image** `<image>`: After processing.

**The task prompt is:** `<TASK_PROMPT>`

**Task Guidelines:**
```
Please determine whether the regions that should remain unchanged between the pre-
and post-processed images exhibit any semantic errors | that is, whether there
are additions, deletions, or modifications of semantic content relative to the
pre-processed image.  If no specific task type is provided, simply determine whether
the two images are completely identical.
```
**Output Format:**
```
Return your decision in a single line of valid JSON with the format:  {"answer":
"Yes", "problem":  "NULL"} if the images are consistent, otherwise {"answer":  "No",
"problem":  ["add", "replace", "remove"]}.  "No" is used whenever any potential
inconsistency is detected.
```

## Format 1: Low-level Appearance

**The first image** `<image>`: Before processing.
**The second image** `<image>`: After processing.

**The task prompt is:** `<TASK_PROMPT>`

**Task Guidelines:**
```
Please determine whether the pre-processed image has undergone any low-level
degradation or shift after processing.  Specifically, check whether there is any
degradation (e.g., blur, noise), color cast, or newly introduced artifacts.  If the
processing described in the task type is explicitly related to any of the above (e.g.,
denoising, deblurring, color correction, artifact removal), then this case should be
ignored.  If no specific task type is given, simply judge whether the two images are
identical.
```
**Output Format:**
```
Return your decision in a single line of valid JSON with the format:  In the "ignored"
case, output:{"answer":  "NULL", "problem":  "NULL"}, {"answer":  "Yes", "problem":
"NULL"} if the images are consistent, otherwise {"answer":  "No", "problem":  ["noise",
"blur", "color cast", "exposure degradation", "artifact"]}.
```

## Format 2: Semantic Level

**The first image** `<image>`: Before processing.
**The second image** `<image>`: After processing.

**The task prompt is:** `<TASK_PROMPT>`
```
Assume this example DOES contain issues:  semantic drift has occurred within regions
that should be preserved.  Drift type(s):  <TYPES_FROM_FORMAT_1_RESULT>
```

**Task Guidelines:**
```
Your task is to analyze the problem strictly in two stages:
1) Preservation analysis (think):  - Identify the intended edit target region(s)
according to the task prompt.  - Explicitly state which changes can be ignored because
they fall inside the intended edit scope.  - Identify the regions/elements that must
be preserved (non-edit regions), and list them as a concrete checklist with brief
justification.
2) Problem reporting (problem):  - Report ONLY issues that violate the preservation
analysis above.  - If something was stated as ignorable or allowed-to-change in the
think stage, it MUST NOT appear here.  - Focus on preserved regions and explain the
semantic drift clearly.  - Use only the drift type keys that were provided above
(Drift type(s):  XXX).
```
**Output Format:**
```
Output MUST be a single valid JSON object and nothing else:  { "think":  "Preservation
analysis:  intended edit target and ignorable changes first; then a checklist of
preserved elements with justification.", "problem":  { "add":  "Describe added content
in preserved regions (if applicable).", "replace":  "Describe replaced content in
preserved regions (if applicable).", "remove":  "Describe removed or missing content
in preserved regions (if applicable)." } }
```

## Format 2: Low-level Appearance

**The first image** `<image>`: Before processing.
**The second image** `<image>`: After processing.

**The task prompt is:** `<TASK_PROMPT>`
```
You are evaluating whether the AFTER image introduces unintended LOW-LEVEL
degradation/shift in regions that should be preserved.  Candidate degradation type(s)
(use ONLY these keys if applicable):  <TYPES_FROM_FORMAT_1_RESULT>
```

**Task Guidelines:**

```
Your task is to analyze strictly in two stages:
1) Preservation & scope analysis (think): - Identify the intended target region(s)
implied by the task prompt. - State which changes are allowed ONLY if they occur
strictly inside the intended target region(s). - Clarify that low-level degradations
(noise/blur/color cast/exposure issues/artifacts) are NOT intended unless the task
prompt explicitly requests low-level enhancement/removal. - Provide a checklist of
what must be preserved (non-target regions/elements) with brief justification. - If
the task prompt explicitly requests low-level processing (e.g., denoise/deblur/color
correction/exposure enhancement/artifact removal), then low-level changes consistent
with that request and confined to the intended scope may be treated as allowed; state
this in "think".
2) Problem reporting (problem): - Report ONLY low-level degradations that violate
the scope above (i.e., occur in preserved regions or exceed intended scope). - If
no violation is found, output an empty object: problem = . - Use ONLY the keys
provided in YYY. Do not invent new keys. - For each key you include, describe: where
it appears, how it differs from BEFORE, and the visual symptom.
```
**Output Format:**
```
Output MUST be a single valid JSON object and nothing else: { "think": "...",
"problem": { "noise": "...", "blur": "...", "color cast": "...", "exposure
degradation": "...", "artifact": "..." } }
```

## B. Supplementary Experimental Results

### B.1. Supplementary Results of StableI2I and MLLMs on StableI2I-Bench

As discussed in the main paper, the input prompt template used by StableI2I at inference time is not identical to the template adopted in StableI2I-Bench for evaluating general MLLMs. This is because StableI2I is a model specifically trained for I2I fidelity assessment, and its training relies on a fixed and unified task template. In contrast, the benchmark template contains richer instructional priors to enable general-purpose MLLMs to correctly perform the evaluation task.

*Table 10.* Quantitative comparison of mainstream models on StableI2I-Bench. Binary Accuracy measures answer correctness, while Strict Accuracy additionally requires correct error types. Best and second-best results are highlighted in **dark blue** and **light blue**, respectively.

| Models | Binary Accuracy | | | | Strict Accuracy | | | |
|---|---|---|---|---|---|---|---|---|
| | Structure | Semantic | Low-level | Avg. | Structure | Semantic | Low-level | Avg. |
| **Open-Source Models** | | | | | | | | |
| Qwen3VL-8B-Instruct | 49.80 | 48.60 | 79.60 | 59.33 | 37.00 | 22.20 | 52.30 | 37.17 |
| Qwen3VL-32B-Instruct | 53.70 | 51.60 | 87.90 | 64.40 | 40.30 | 24.20 | 58.50 | 41.00 |
| InternVL-3.5-8B | 38.40 | 46.90 | 59.10 | 48.13 | 28.20 | 6.20 | 14.60 | 16.33 |
| InternVL-3.5-38B | 41.30 | 46.70 | 75.50 | 54.50 | 24.60 | 18.60 | 40.30 | 27.83 |
| **Proprietary Models** | | | | | | | | |
| Grok-4.1 | 57.70 | 58.90 | 73.70 | 63.43 | 51.80 | 39.10 | 40.30 | 43.73 |
| Claude-Sonnet-4.5 | 65.00 | 70.40 | 88.70 | 74.70 | 61.60 | 53.10 | 71.20 | 61.97 |
| Claude-Sonnet-4.5-think | 65.40 | 66.50 | 86.30 | 72.73 | 62.90 | 51.90 | 68.60 | 61.13 |
| Gemini-2.5-pro | 68.17 | 77.50 | 92.50 | 79.39 | 63.36 | 55.30 | 60.10 | 59.59 |
| Gemini-3-pro | 70.77 | 80.14 | 83.98 | 78.30 | 64.26 | 59.17 | 61.06 | 61.50 |
| GPT-4o | 58.50 | 80.10 | 88.10 | 75.57 | 53.90 | 60.30 | 76.30 | 63.50 |
| GPT-5 | 66.20 | 82.50 | 95.20 | 81.30 | 61.00 | 64.10 | 63.20 | 62.77 |
| StableI2I | 85.40 | 82.80 | 99.10 | 89.10 | 83.70 | 67.30 | 98.00 | 83.00 |

To provide a more controlled and fair comparison, we further conduct an additional experiment in which MLLMs are evaluated using the same input template as StableI2I. This setting allows us to isolate the effect of the prompt template and to better assess the intrinsic performance gap between StableI2I and general MLLMs under identical prompting conditions. The comparative results are shown in Tab. 10.

We observe that, compared with Tab. 1 in the main paper, the performance of MLLMs drops the most on the Semantic Level dimension when the additional instructional priors are removed. In contrast, the performance on the Structure Level remains largely unchanged and even exhibits a slight improvement. We attribute this behavior to the fact that the Structure Level is inherently difficult for current models to judge. As a result, the presence or absence of additional priors does not substantially help models achieve reliable performance on this dimension.

Importantly, even under this significantly less informative prompting condition, StableI2I consistently outperforms all general-purpose MLLMs across all three evaluation dimensions. This result indicates that the superior performance of StableI2I does not stem from a more favorable or information-rich prompt design, but rather from its task-specific training and its enhanced pixel-level perceptual and alignment capabilities.

Moreover, this controlled comparison confirms that the benchmark prompt template does not confer an unfair advantage to StableI2I. On the contrary, the instruction-rich template in StableI2I-Bench primarily serves to make the evaluation task feasible for general-purpose MLLMs, rather than to artificially inflate their performance. When this auxiliary instructional prior is removed, MLLMs exhibit substantial performance degradation, whereas StableI2I remains robust and maintains strong performance across all dimensions.

These findings collectively demonstrate that the performance gap between StableI2I and general MLLMs reflects an intrinsic capability difference, rather than a confounding effect introduced by prompt engineering. This further validates the necessity of a task-specialized fidelity assessment model and highlights the limitations of current general-purpose MLLMs in fine-grained I2I fidelity evaluation.

### B.2. Ablation study with ImgEdit-Judge

Since the two models adopt different evaluation dimensions, it is difficult to directly assess single-image results. Therefore, we compare the outputs produced by two different models and evaluate whether human preference judgments are consistent with the corresponding evaluation results. Specifically, ImgEdit-Judge is evaluated using the *Physical & Detail Integrity* dimension. Experiments are conducted on two benchmarks, **ImgEdit-Bench** and **GEdit-Bench**. For each benchmark, we randomly sample 50 comparison pairs from a candidate pool consisting of Qwen-Image-Edit-2511 (Wu et al., 2025a), Nano-Banana (Google, 2025), GPT-Image-1 (OpenAI, 2025b), and Omnigen2 (Wu et al., 2025b). The final results are reported in Tab. 11, showing that our model achieves better performance than ImgEdit-Judge in fidelity assessment.

*Table 11.* Comparison of fidelity assessment accuracy between ImgEdit-Judge and StableI2I on ImgEdit-Bench and GEdit-Bench.

| Accuracy | ImgEdit-Bench | GEdit-Bench |
|---|---|---|
| ImgEdit-Judge | 30% | 52% |
| StableI2I | 80% | 76% |

### B.3. Supplementary Results of StableI2I in Real-World I2I Evaluation Settings

To better illustrate the evaluation performance of our model, we present additional subjective result examples. Specifically, the results on ImgEdit-Bench are shown in Fig. 12, those on GEdit-Bench are shown in Fig. 13, and the results on the Low-level Dataset are shown in Fig. 14.

The results presented above correspond to the model's short-form answers to the evaluation questions. Fig. 15 further illustrates detailed descriptions generated by the model for erroneous cases along the Semantic Level and Low-level Appearance dimensions. In each example pair, the left image is the input image and the right image is the output image. The model is able to accurately identify and describe both the error types and the corresponding target objects.

### B.4. Demonstration of Model Limitations

In our experiments, we also observed several failure cases, as shown in Fig. 11. For example, in the case of style transfer, the style change itself constitutes a form of content repainting. From a human perspective, such an edit should be considered valid and correct; however, the model still classifies it as problematic repainting. For object extraction tasks, the model may misjudge the result as losing other content, which can be regarded as a failure case that lies outside the distribution of the training data. In addition, for human pose change scenarios, such edits inevitably cause pixel-level misalignment in other parts of the human body. Nevertheless, from a semantic and task-oriented perspective, these editing results are correct.

In summary, the errors made by the current model mainly stem from cases where the actual I2I processing direction overlaps or conflicts with the model's judgment direction. In the future, we plan to expand the scope of I2I tasks to better cover such cases, thereby enhancing the model's perceptual capability for complex editing semantics. Moreover, in such ambiguous situations, allowing the model to selectively abstain from answering can also be a reasonable solution. We also

believe that these issues are, to some extent, attributable to limitations in model capacity and parameter scale. In contrast, closed-source large models usually exhibit more flexible adaptation capabilities and tend to perform better in these scenarios. If closed-source multimodal large models could achieve pixel-level perceptual sensitivity comparable to that of StableI2I, we believe that multimodal large models as a whole would make further progress.

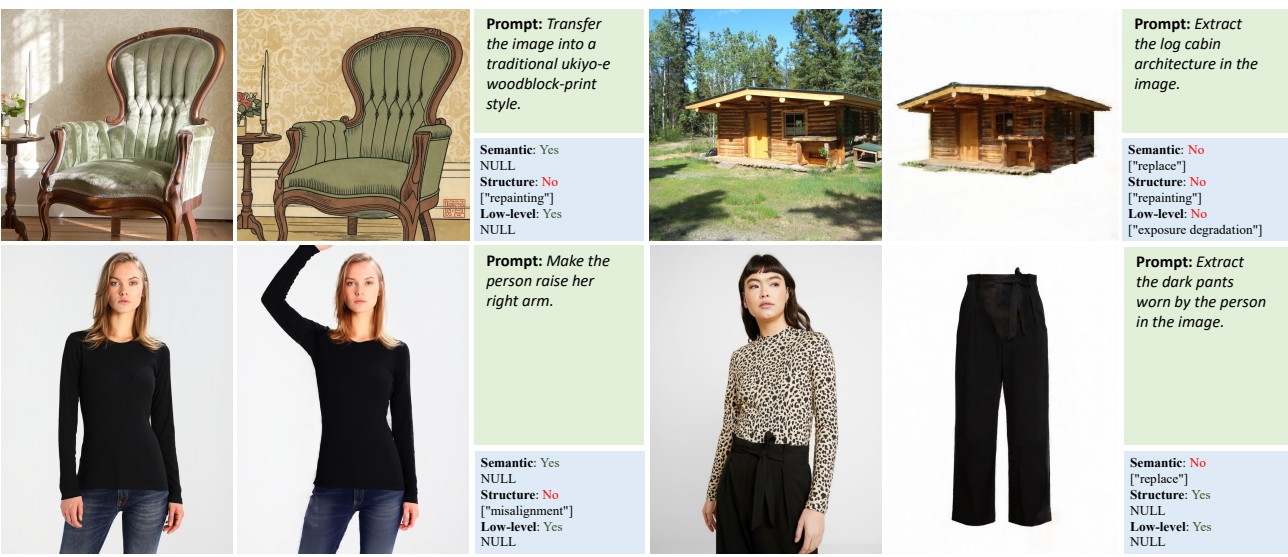

*Figure 11.* The results shown above correspond to **failure cases** of StableI2I, reflecting its limitations. Under human judgment, all of these tasks should be considered correct; however, the model fails to complete them successfully. The specific editing types, from the top left to the bottom right, are: style transfer, object extraction, human motion, and object extraction.

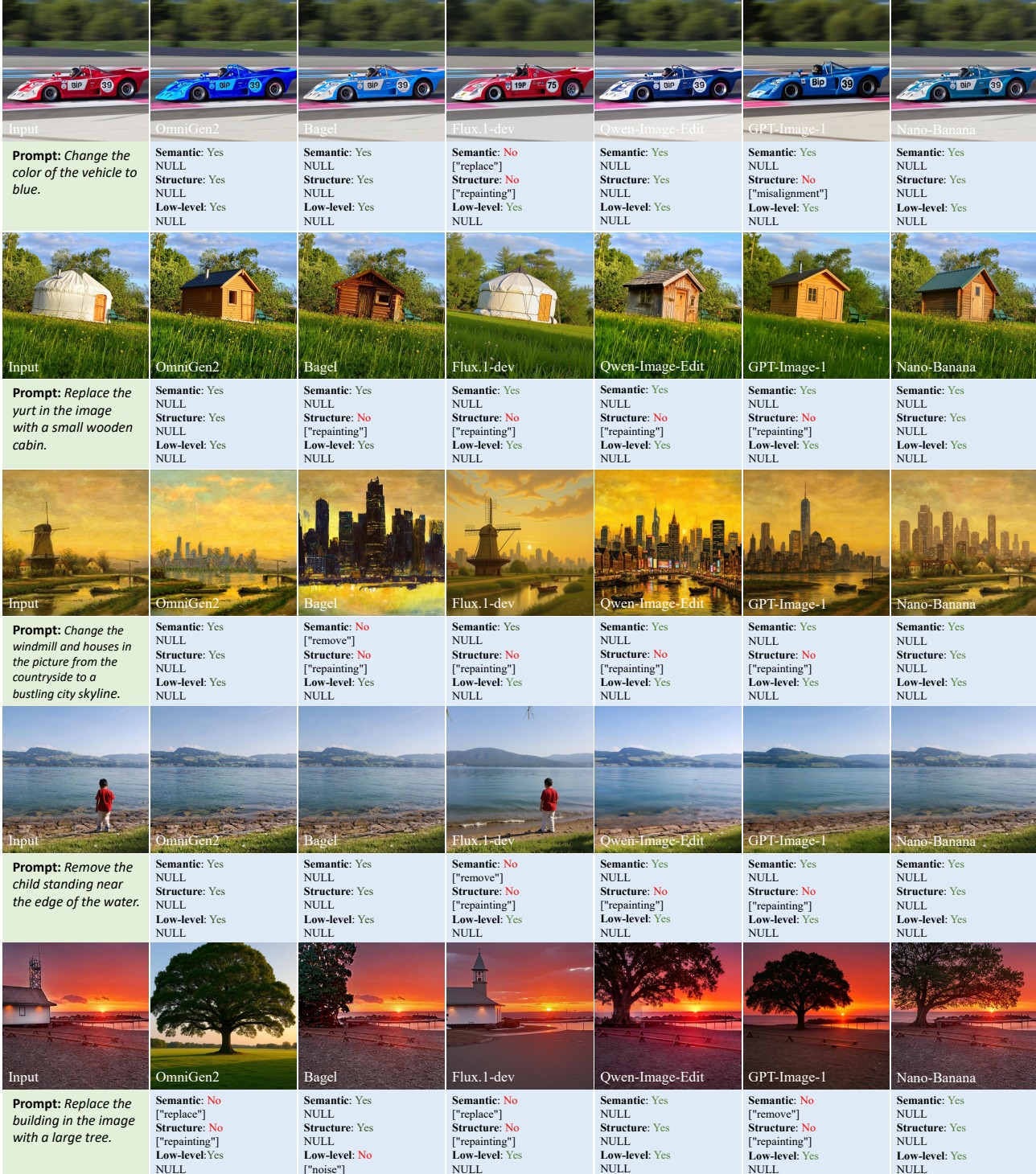

*Figure 12.* Visualization of the evaluation results on ImgEdit-Bench using Format 1.

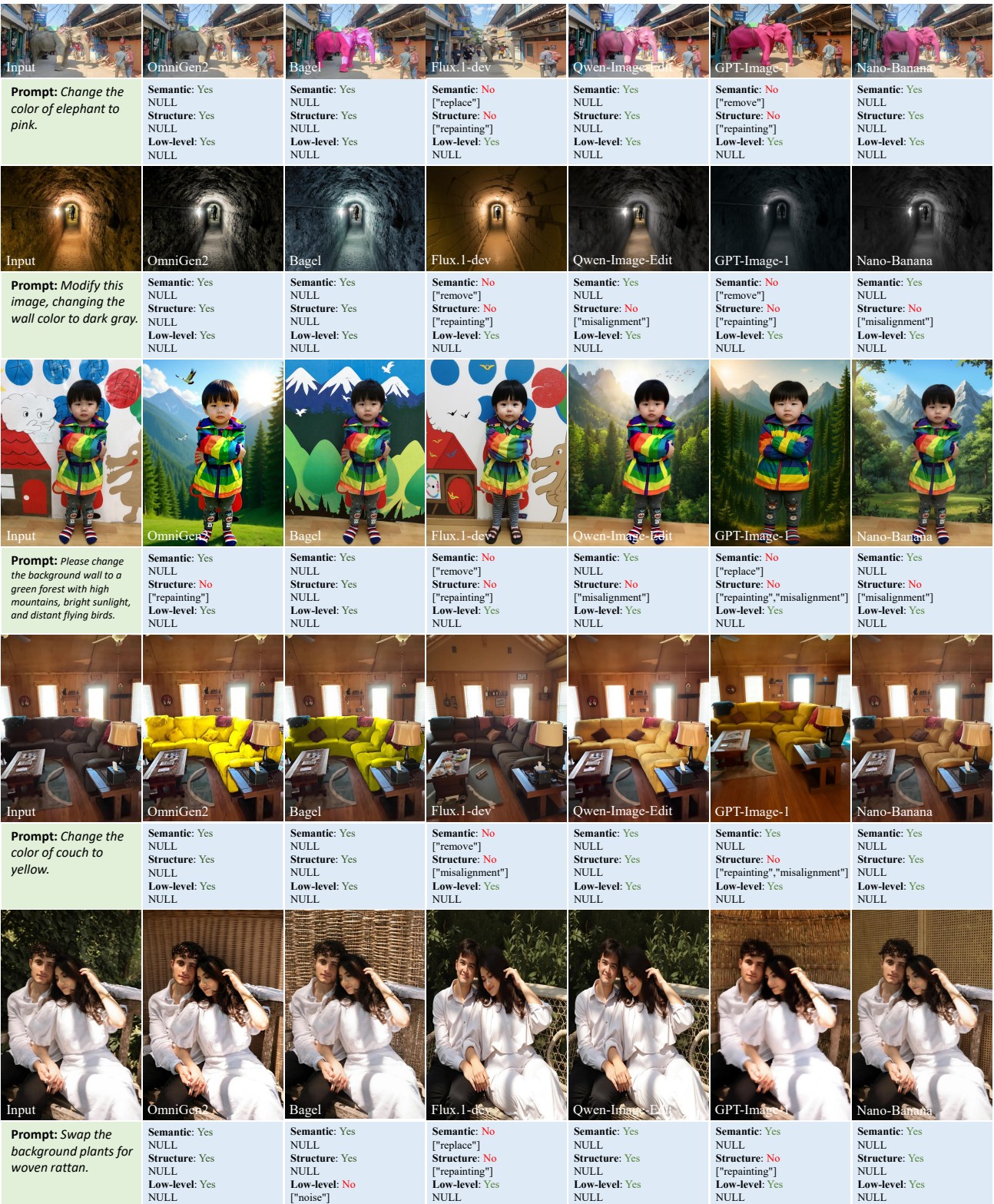

Figure 13. Visualization of the evaluation results on GEdit-Bench using Format 1.

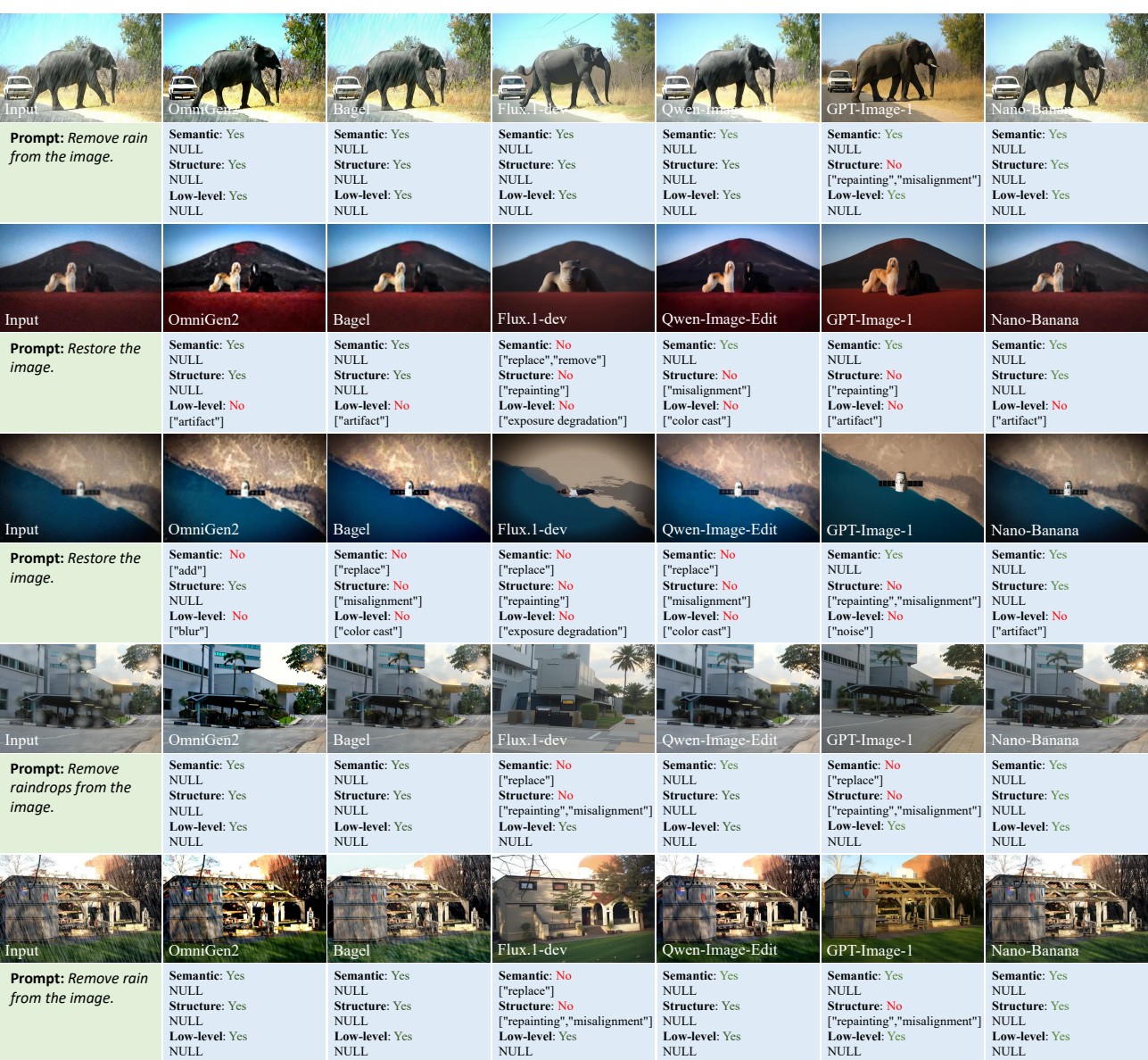

*Figure 14.* Visualization of the evaluation results on Low-level Dataset using Format 1.

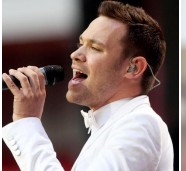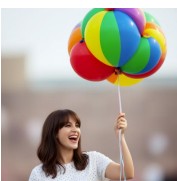

**Think**: *Check that the background and framing remain consistent while ignoring the subject replacement and balloon addition.*
**Problem**:
"replace": Background changed from red stadium-like setting to a neutral outdoor scene.

**Prompt:** Replace the singer in the image with a person holding a colorful balloon.

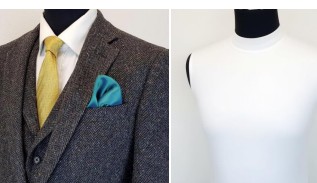

**Think**: *Check shirt, pocket square, and mannequin remain; ignore jacket/tie removal and related minor changes.*
**Problem**:
"replace": White dress shirt and mannequin neck changed to a high-neck mock turtleneck.

**Prompt:** Remove the suit jacket and tie combination in the image.

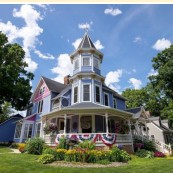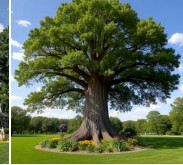

**Think**: *Check lawn, sky, and background trees remain; ignore the house-to-tree replacement and its local effects.*
**Problem**:
"remove": Front porch and surrounding flowerbeds are missing.

**Prompt:** Replace the house in the image with a giant tree.

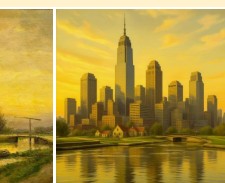

**Think**: *Check that the river, boats, bridge, and rural landscape remain; ignore the skyline replacement itself.*
**Problem**:
"remove": Boats on the river and the small rural bridge are missing.

**Prompt:** Change the windmill and houses in the picture from the countryside to a bustling city skyline.

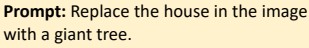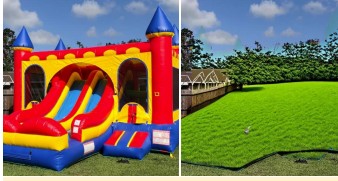

**Think**: *Check preservation of the rest of the park, trees, sky, and distant structures while ignoring the removed bounce house and its immediate area.*
**Problem**:
"replace": Background park/trees and sky were transformed into a uniform green lawn and altered colors, changing non-target scenery.

**Prompt:** Remove the colorful inflatable bounce house with slides in the foreground.

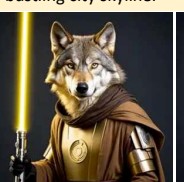

**Think**: *Check that the costume, lightsaber, and background are preserved; ignore the wolf→bear change and bear paws on the lightsaber.*
**Problem**:
"remove": The brown costume and armor from the original were removed.

**Prompt:** Replace the wolf with a standing bear, and have the bear hold a lightsaber with both paws.

**Think**: *Even if the semantic edit is correct, the output is invalid if it introduces low-level quality regressions in preserved regions. Therefore, we focus on objective low-level cues: edge sharpness, texture clarity, noise granularity, global/local brightness shifts, chromatic bias, and compression/processing artifacts.*
**Problem**:
"color cast": Cool color cast (medium).

**Prompt:** Change the background to a starry sky.

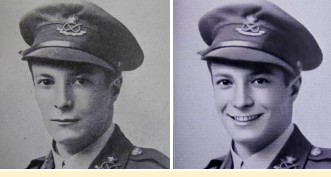

**Think**: *Preservation principle: Non-target content should remain semantically identical. Low-level degradations (noise, blur, color cast, exposure degradation, artifacts) are not part of normal semantic edits and must not be introduced in preserved regions.*
**Problem**:
"color cast": Cool color cast (medium).

**Prompt:** Make the person in the image smile.

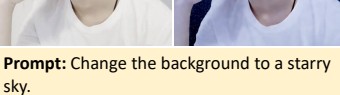

**Think**: *Check lawn, sky, and background trees remain; ignore the house-to-tree replacement and its local effects.*
**Problem**:
"remove": Front porch and surrounding flowerbeds are missing.

**Prompt:** Replace the house in the image with a giant tree.

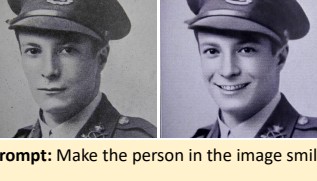

**Think**: *Check that the river, boats, bridge, and rural landscape remain; ignore the skyline replacement itself.*
**Problem**:
"remove": Boats on the river and the small rural bridge are missing.

**Prompt:** Change the windmill and houses in the picture from the countryside to a bustling city skyline.

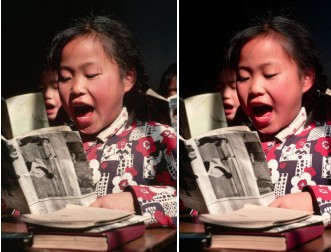

**Think**: *Check that the girl, background, and especially the book pages remain the same while ignoring lighting and minor framing changes.*
**Problem**:
"replace": Book page grayscale photo changed to a different person/image compared to the original.

**Prompt:** Enhance the image quality.

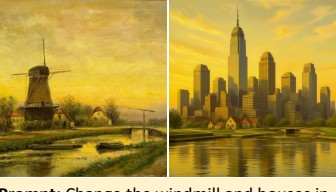

**Think**: *Check that the lighthouse, figure, and field remain the same while ignoring the removed sky papers and tonal changes.*
**Problem**:
"remove": Floating paper fragments in the sky are missing.

**Prompt:** Restore the image.

*Figure 15.* Illustration of detailed results answered using Format 2.

