# OpenReview forum: "StableI2I: Spotting Unintended Changes in Image-to-Image Transition"
_ICML.cc/2026/Conference — ICML 2026 regular_

### Official Review · Reviewer_ra2o · 2026-03-02

**Soundness:** 3
**Presentation:** 3
**Significance:** 3
**Originality:** 3
**Overall Recommendation:** 5
**Confidence:** 4

**Summary:**

This paper proposes StableI2I, an unrefereed evaluation framework and dedicated evaluation model for image-to-image (I2I) editing and restoration scenarios. The core idea is to split the content fidelity from the input image and instruction conditions into three dimensions: Semantic, Structure, and Low-level Appearance, and output binary and error type diagnoses. At the same time, StableI2I-Bench is constructed for system evaluation of MLLM's ability in multi-image consistency and fine-grained perception. Through multi-stage data construction, SFT, GRPO, self-labeling and GPT-5 filtering of Open-ended QA, a 8B dedicated judge is trained. Experimental results show that it significantly outperforms general MLLM on the benchmark and has higher consistency with human judgment.

**Compliance With Llm Reviewing Policy:**

Affirmed.

**Final Justification:**

Thank you for the authors' rebuttal. The clarifications addressed some of my concerns and I have updated my score from 4 to 5 accordingly.

**Key Questions For Authors:**

1. How much of StableI2I's performance comes from the GPT-5 filtering and generating open-ended QA? Please provide the ablation curves for step-by-step removal and the changes in failure types.

**Limitations:**

yes

**Strengths And Weaknesses:**

Strengths
1. The problem definition precisely targets the real pain points. The existing I2I evaluations often focus on aesthetics or coarse semantic consistency, while neglecting "unexpected drift in non-target areas". The paper sets this as its primary goal and emphasizes the needs of high risk applications, and the motivation is valid.
2. The three dimensional diagnosis output is relatively operable. Drift is broken down into semantic mismatch, structural misalignment and repainting, as well as low-level noise, etc.
3. It provides a relatively complete set of training details and statistics on data scale, including GRPO training settings, the amount of editing and restoration data, and the manual cleaning process. The cost and feasibility are clear.

Weaknesses
1. The evaluation protocol has significant "prompt prior" confounding, and fairness and comparability are not established. The paper acknowledges that the StableI2I inference template is different from the StableI2I-Bench template for the general MLLM, and that the general model will significantly drop scores after removing the structured prior of the bench template.
2. The method involves non-reusable algorithm innovations and is overly dependent on GPT-5, which affects reproducibility. The key data construction and annotation links heavily rely on GPT-5, making it impossible for others to reproduce the results, and the model may have learned GPT-5's preferences rather than objective consistency criteria.

---

> ### Author Rebuttal · Authors · 2026-03-30
>
> **W1: On Prompt Fairness**  Thank you for your question. We did indeed consider this issue, which is why we included supplementary experiments in both the main paper and the appendix. First, our model is a specialized model, so unlike general-purpose models, it cannot rely on flexible prompt control in the same way. Therefore, its prompt format is aligned with its training setup.
>
> The results of applying the same prompt as StableI2I to other models are presented in Appendix B.1, *Supplementary Results of StableI2I and MLLMs on StableI2I-Bench*. The prompt used there already contains some simple prior information. To ensure fairness, we further injected more prior information into the prompts given to the other models, while StableI2I still retained its original prompt format; these are the results reported in the table in the main paper.
>
> Although the performance of general-purpose models improves compared with using the earlier raw prompts, they still lag far far behind the specialized model StableI2I. Since most existing evaluations rely on closed-source models together with carefully designed prompts, these results also clearly show that even the strongest closed-source models, when paired with prompting alone, still struggle to make such specialized judgments well. This further demonstrates that dedicated expert models are still necessary for evaluation.
>
> **W2: On GPT-5 Dependency and Reproducibility**  We clarify that GPT-5’s preferences do not play a dominant role in our pipeline. In our data construction, GPT-5 is mainly used for semantic-level verification, i.e., judging whether preserved content has been incorrectly altered. This provides a correctness-based signal rather than a preference-based one.
>
> More importantly, GPT-5 itself performs poorly on our structure-level, pixel-sensitive perception tasks, as shown in the benchmark results. This suggests that our model is not merely inheriting GPT-5’s preferences or capabilities. For structure-level supervision, we rely only on human-annotated data and self-supervised synthetic augmentation, without using any large model annotation. The capability upper bound is further improved during RL with human-annotated data, so the main supervisory signal still comes from human annotations.
>
> We also agree that Open-ended QA responses may show some stylistic traces of GPT-5 or other strong models. However, our focus is not the writing style, but whether the fine-grained analysis is accurate. From this perspective, using strong models to generate or examine such analyses is reasonable.
>
> **Q1: On Component Contribution and Ablation**  Thank you very much for your question. Let us first briefly explain the progression from StableI2I-Dev.2 to the final version. As stated in the main paper, the Dev.2 version can significantly improve performance on the Semantic Level, but it also leads to a decline in Structure Level performance. Therefore, our goal was to further enhance SFT data in a bootstrapping manner, so as to strengthen the model’s semantic capability while avoiding degradation in structure-level performance.
>
> To achieve this, we first used Dev.2 to annotate a large amount of data, and then provided these annotated answers to GPT-5 as hints to judge whether they were accurate, thereby filtering out annotation pairs with obvious errors. Since the problematic data have already been removed, we are unfortunately unable to provide an ablation study specifically isolating the effect of this filtering step.
>
> As for Open-ended QA, it can be understood as a more refined reflection process rather than simply producing a direct answer to a question. Therefore, it improves performance on both the Semantic Level and Low-level Appearance. The ablation results for Open-ended QA are shown in the table below, where all numbers are based on Strict Accuracy on StableI2I-Bench.
>
> | Setting | Structure Level | Semantic Level | Low-level Appearance |
> |---|---:|---:|---:|
> | Base | 53.5 | 21.4 | 24.6 |
> | StableI2I-Dev.1 (early version) | 81.9 | 54.4 | 94.1 |
> | + Filter data | 81.4 | 65.3 | 97.7 |
> | + Filter data + Open-ended QA | 81.2 | 66.1 | 98.0 |
>
> It is worth noting that the results in this version were preserved from our early validation experiments, so the numbers for StableI2I-Dev.1 are not the same as those reported for the final version in the paper. The first row corresponds to the base capability of Qwen3-VL-8B-Instruct; the second row shows the results of StableI2I-Dev.1; the third row adds filtered data on top of the training data used for StableI2I-Dev.1; and the fourth row further adds Open-ended QA on top of the third setting.
>
> We can observe that, as these components are progressively added, both Semantic Level and Low-level Appearance improve. Since the improvement of Structure Level is more closely tied to Texture-Aware Enhancement Data, its performance declines somewhat when the proportion of other types of data increases.

---

> > ### Author Rebuttal · Reviewer_ra2o · 2026-04-01
> >
> > Thank you for the response and additional experiments. The clarifications addressed some of my concerns and I have updated my score accordingly.

---

> > > ### Author Response · Authors · 2026-04-01
> > >
> > > Thank you very much for your positive update. We are glad that our response and additional experiments helped address your concerns, and we sincerely appreciate your time and consideration.

---

### Official Review · Reviewer_q7hA · 2026-03-12

**Soundness:** 2
**Presentation:** 4
**Significance:** 2
**Originality:** 2
**Overall Recommendation:** 4
**Confidence:** 5

**Summary:**

The motivation of the paper is that existing evaluations of Image-to-Image (I2I) tasks mostly focus on instruction following and image quality, while neglecting the semantic and structural consistency between inputs and outputs, which is prone to causing problems in high-stakes scenarios. The core work is proposing the StableI2I reference-free evaluation framework, which measures fidelity from three dimensions—semantic level, structure level, and low-level appearance—constructing the StableI2I-Bench benchmark and a multi-stage data pipeline, and training the model with multiple types of data. The main contributions are filling the gap in I2I fidelity evaluation, providing an accurate and interpretable evaluation tool, offering a standardized benchmark for MLLM-related evaluations, and helping improve the reliability of I2I systems.

**Compliance With Llm Reviewing Policy:**

Affirmed.

**Final Justification:**

The authors have addressed my concerns in the rebuttal. Based on this, I increase my score to 4.

**Key Questions For Authors:**

Please explain the reasons for the counterintuitive evaluation results, specifically referring to the second point in the weaknesses section.

**Limitations:**

yes

**Strengths And Weaknesses:**

Strengths:
1. The paper has a clear motivation: existing Image-to-Image (I2I) evaluation methods mainly focus on the aesthetic quality, perceptual metrics, and instruction-following ability of generated images, while neglecting the semantic correspondence and spatial structural consistency between inputs and outputs.
2. The benchmark designed in the paper defines three fidelity dimensions—Semantic Level, Structure Level, and Low-level Appearance—with data sourced from over 11,000 annotated data pairs generated through a combination of model-assisted and manual annotation.
3. The paper is clearly written, with relatively comprehensive experiments.

Weaknesses:
1. Insufficient coverage of benchmark tasks. The image editing tasks only involve three basic operations (add, replace, and remove), failing to cover tasks such as style transfer and pose modification, as well as more complex image editing scenarios. Additionally, modern image editing models are not limited to evaluating cases where the input only contains a single image.
2. Doubts about the validity of the proposed benchmark. Evaluation results show the performance ranking as Bagel > Qwen-Image-Edit2509 > Qwen-Image-Edit2511. Furthermore, Nano-Banana performs worse than other models on many metrics. As a deep user of these models, this is seriously inconsistent with practical experience.
3. Lack of technical depth, making it less suitable for the ICML conference.

Overall, this paper addresses an important problem, but the current solution is incomplete and cannot fully evaluate the performance of I2I models.

---

> ### Author Rebuttal · Authors · 2026-03-30
>
> **W1: Clarification on task coverage.** Thank you very much for your question. We would first like to clarify a key misunderstanding. Our benchmark does not restrict image editing tasks to only three operations. Under our predefined prompt framework, all editing instructions are randomly generated by GPT-5, and the resulting task set covers a broad range of edits, including style transfer, pose modification, brightness adjustment, season transformation, daytime change, and other realistic editing scenarios.
>
> What is limited to three categories is **only the Semantic Level evaluation protocol**, where we check whether the output image introduces semantic errors in terms of add, replace, or remove of content. This should not be confused with the diversity of the editing tasks themselves. For example, for a daytime-change instruction, the intended semantic content of the scene should remain unchanged, so the semantic evaluation naturally falls into the preservation setting rather than a new semantic edit category.
>
> We agree that the current benchmark focuses on single-image I2I settings and does not yet cover multi-image editing. We choose this setting because single-image editing remains the most widely studied, standardized, and mature setup in current research, and is also adopted by representative benchmarks such as ImgEdit, making it a suitable starting point for controlled and fair evaluation. More importantly, our experiments show that even in this canonical setting, current state-of-the-art models still exhibit substantial fidelity issues, especially in preserving regions that should remain unchanged. We therefore believe it is both necessary and meaningful to first study this problem in the single-image setting, while viewing multi-image and more compositional scenarios as important future extensions rather than evidence against the validity of the current benchmark.
>
> **W2, Q1: Clarification on benchmark validity.** We are very glad that you raised this point, because it touches exactly on the distinction our work is trying to make. The rankings you mentioned may appear counterintuitive if one interprets our score as an overall image quality or task completion score. However, that is not what StableI2I is designed to evaluate.
>
> Our method focuses specifically on fidelity, namely whether the image regions that are supposed to remain unchanged are in fact preserved faithfully after editing. This is different from evaluating instruction following, semantic completion, or overall perceptual attractiveness. As a result, a model can receive a high fidelity score even if its final output is less impressive in terms of edit quality, as long as it preserves the non-target regions well. Conversely, a model that is stronger at following instructions or generating visually appealing results may still receive a lower fidelity score if it introduces unnecessary drift in regions that should have remained unchanged. We provide a concrete example of this phenomenon in Figure: https://anonymous.4open.science/r/anonymous-icml-2026-StableI2I/R3.pdf.
>
> This distinction also helps explain the ranking behavior. For example, **Bagel** sometimes produces results that resemble mask-based compositing. While such outputs may not always look the most natural overall, they often preserve untouched regions extremely well, which is exactly what our fidelity-oriented evaluation rewards. Similarly, the **Nano-Banana** result discussed by the reviewer refers to the earlier Gemini release rather than the stronger **Nano-Banana Pro** version that is more widely used today, so the reviewer’s practical impression may not match the specific model version evaluated in our paper. One possible reason why the **Qwen** series receives relatively low scores is that it may have been trained on a large amount of GPT-image-generated data, causing Qwen in some cases to produce images with a style similar to GPT-image.
>
> More broadly, we believe these seemingly counterintuitive cases highlight the need for our benchmark. Existing evaluations often conflate instruction following, aesthetics, and fidelity, making it hard to isolate whether non-edited content is truly preserved. Our benchmark is designed for this underexplored but practically important dimension.
>
> **W3: Clarification on technical depth.** Thank you for this comment. We would like to clarify that the technical depth of our work lies in identifying and formalizing a previously underexplored problem in image editing evaluation: fine-grained fidelity assessment for I2I tasks. To the best of our knowledge, this is ***the first work dedicated to this problem*** and the first to build dedicated training data and benchmark sets for fidelity evaluation. Based on this, we ***develop a specialized evaluator*** that effectively measures whether non-target regions are faithfully preserved after editing. We believe this establishes a new and important evaluation dimension for current I2I assessment.

---

> > ### Author Rebuttal · Reviewer_q7hA · 2026-04-01
> >
> > I would like to thank the authors for the response.  The authors have addressed my concerns in the rebuttal. Based on this, I will increase my score to 4.

---

> > > ### Author Response · Authors · 2026-04-01
> > >
> > > Thank you very much for your thoughtful follow-up and for updating your score. We are glad that our rebuttal helped address your concerns, and we sincerely appreciate your time and consideration.

---

### Official Review · Reviewer_BQbe · 2026-03-12

**Soundness:** 2
**Presentation:** 2
**Significance:** 3
**Originality:** 2
**Overall Recommendation:** 4
**Confidence:** 4

**Summary:**

Evaluating image-to-image (I2I) generative models that are used typically for image-editing and image restoration tasks is a challenging problem - with existing methods focussing mostly on the textual instruction alignment and perceptual quality metrics. However, these methods still lack a consistent evaluation of key image attributes - semantic, structural and pixel consistency (all at the same time) which might give a misleading picture of the model’s faithfulness to the task - pre vs post I2I transition. In this work the authors aim to solve this problem by proposing the following contributions:

1. An evaluation model for I2I tasks  - StableI2I that is capable of jointly evaluating semantic-level and pixel-level consistency.
2. An evaluation benchmark - StableI2I-Benchmark to evaluate the high-level and low-level visual reasoning abilities of I2I models for fidelity evaluation.
3. They evaluate the proposed model and benchmark - across several open-source and closed-source MLLMs - and show the utility of their method.
4. Their proposed model StableI2I seems to consistently outperform off-the shelf models.

**Compliance With Llm Reviewing Policy:**

Affirmed.

**Final Justification:**

Thanks very much to the authors to respond to my concerns, they more or less have been resolved. However, since many of the changes have to be integrated into the final version which is not visible yet - I would strongly recommend to the authors to integrate all of these into the updated manuscript.

Rest I will keep my positive rating of 4 (WA). This is for two reasons:

1. In its current state the manuscript is not perfect, and heavily relies on the appendix. The authors justification is lack of space - but well that is something that is part of the submission guidelines and has to be followed by all authors - thus it is not fully fair to rely on this reason.
2. The reproducibility concerns have been attended to (by providing an inference code) and the clarification regarding the need for human annotation and auxiliary data has also been somewhat addressed, there are still some reservations about the full pipeline being somewhat handcrafted and not yet completely verifiable during the review process.

**Key Questions For Authors:**

1. Could the authors explain the texture aware enhancement data part more clearly?
2. How does this method compare to the metrics mentioned in Figure 1 ? atleast in terms of consistency ?

**Limitations:**

Yes, the limitations are discussed. However it would also help to highlight the training challenges, dataset collection and reliance on human annotations in the limitations section ? Also it would be nice to see an experiment where one can show what minimal amount of human annotations one needs and how the performance scales with more human annotations for the RL training part.

**Strengths And Weaknesses:**

The following are the strengths of the work:

1. Presentation: The work is well structured, with decent illustrations, figures and tables.
2. Significance: The work is well motivated. It tackles and aims to solve an important existing challenge in evaluating the visual properties and faithfulness of image-to-image transition models, which is becoming increasingly important in today’s landscape of these Multi-Modal Large Language Models becoming more and more prevalent.
3. Originality: The work proposes a framework that leverages existing foundational models, and a human annotation pipeline - to generate a training + evaluation benchmark StableI2I-Bench and train a model StableI2I that is then used to evaluate the semantic, structural and pixel wise consistency of the image to image transitions that the models perform. The pipeline is quite interesting and the benchmark could be potentially useful.
4. Soundness: The experiments do show comparisons for a large suite of open-source and closed models and demonstrate the utility of the proposed model and benchmark.

The following are the weaknesses of the work:

1. Presentation: The work although well-structured, the writing and explanation could be improved especially for the Main Method section. The Figure2. is not very clear and informative (including the captions) - ideally the figures + captions should be self-sufficient, but here even the supporting text is not too clear. Since this is the main figure it should be made more apparent what is actually going on, especially for the Multi-type Degradation dataset part.
2. Originality: The main method relies heavily on ChatGPT API that is a state-of-the-art closed foundation model to clean and generate annotations - and also relies on human annotations (when the motivation says that existing methods like with Full-Reference (FR) have drawbacks since they rely on ground-truths, but this method also relies on human annotations to get the dataset and even training with RLHF objectives). Thus, the originality and significance of the underlying method could be questioned.
3. Soundness: The method section is not very well explained, and heavily references to the Appendix. For the method and key components atleast it should be self-sufficient in the main paper. For instance the Texture-Aware Enhancement Data part is not well explained.
There are missing experiments to show how the proposed method (StableI2I) performs against the metrics used in Figure 1 -  CLIP-IQA, MANIQA, MUSIQ. ArtiMuse, ImgEditJudge. It is understandable that they may not be directly comparable - but having a proxy or some evaluation to exactly quantify how good or bad are these quantitatively vs the proposed method is important.

Additionally to make this method really useful and reproducible - making the datasets used for training and evaluation is essential.

While the motivation of the work is good, the overall pipeline still seems quite handcrafted which still doesn’t really convince the reviewer that this should be the way forward to evaluate such I2I models / tasks.

Minor: I would suggest removing the emoji's and hyperbole that is used in Figure 1: to exaggerate the proposed method. This doesn't seem to be very scientific and professional for a manuscript submitted to a top-tier conference.


Note for rebuttal: If attended to the weaknesses and queries well, the scores could be improved to 5 Accept.

---

> ### Author Rebuttal · Authors · 2026-03-30
>
> **W1: Presentation issue.** Thank you for your feedback. We agree that Fig. 2 and its caption can be clearer, and we will revise them in the final version, especially for the Multi-type Degradation Dataset part. The figure is intended to show that restoration data combines an augmentation-based synthetic pipeline with annotations on degradation cases from real model outputs, while editing data is annotated entirely from degradation cases in real model outputs. “SFT” and “RL” indicate the training stage where the data is used. More details are provided in Appendix *A.1, Data Construction and Statistics*.
>
> **W2: Originality concern.** Thank you for your question. Here, the absence of GT means that, in image-to-image (I2I) tasks, there is usually no single GT output image for evaluation; otherwise, full-reference metrics such as PSNR could be directly applied. In our setting, human or GPT annotations are used only to train the model to judge whether the edited image introduces information errors relative to the input image. Once trained, the model no longer relies on any ground-truth output image, but evaluates the result based on the input itself. This is fundamentally different from GT-based evaluation. Moreover, I2I editing often involves substantial content changes whose reasonableness cannot be captured well by simple pixel-level or mask-based comparisons alone. For example, an edited region may naturally require corresponding changes in lighting or shadows, which mask-based methods often handle too crudely. Human preference alignment is therefore better suited to this setting.
>
> **W3, Q1, Q2:** Thank you for your questions.
>
> **Q1: What is the role of Texture-Aware Enhancement Data?**
> Existing models often preserve the target semantics but still make pixel-level errors, mainly content repainting and pixel-structure misalignment, as shown in our figure：https://anonymous.4open.science/r/anonymous-icml-2026-StableI2I/R2.pdf.
>
> To address this, we first built structure-level training data. However, since this fine-grained perceptual task remains challenging for current models, the data had to be manually annotated by experts. In total, we collected fewer than 10K training images and about 1K test images, which is still limited for learning robust pixel-level perception.
>
> This motivates our Texture-Aware Enhancement Data. The key idea is to randomly transform a natural image into a pair of images from the same scene with subtle misalignment, thereby simulating pixel-level mismatch without manual labeling. This synthetic data helps the model learn fine-grained perceptual sensitivity beyond what limited labeled data alone can provide.
>
> The results below show its effectiveness on the Structure Level of StableI2I-Bench. The three numbers correspond to the base model, training with labeled data only, and training with labeled data plus Texture-Aware Enhancement Data. Labeled data improves performance over the base model, and the enhancement data brings a further substantial gain.
>
> | Setting | Structure-Level Score |
> |---|---:|
> | Base | 0.6377 |
> | + Labeled Data | 0.6822 |
> | + Labeled Data + Enhancement Data | 0.7860 |
>
> **Q2: How does the method compare with the metrics in Figure 1?**
> To ensure a fair comparison, we designed a dedicated pairwise preference experiment. We randomly collected outputs from multiple generative models on ImgEdit and constructed 120 image pairs. After removing problematic samples, 112 valid pairs remained.
>
> Human annotators were then asked to judge which output had higher fidelity. If a metric’s ranking agreed with human preference, we counted it as correct. The results are shown below. Among the compared methods, ImgEdit-Judge is the most relevant baseline because it is also designed for consistency evaluation. As shown in the table, StableI2I achieves substantially higher agreement with human preference than all other metrics.
>
> | Method | ACC |
> |---|---:|
> | CLIP-IQA | 33.04% |
> | MANIQA | 39.29% |
> | MUSIQ | 35.71% |
> | ArtiMuse | 48.21% |
> | ImgEdit-Judge | 32.69% |
> | StableI2I (ours) | 75.89% |
>
> **W4: On Reproducibility**. We agree that data release is important for reproducibility and fair comparison. We will release the human-annotated training data, evaluation set, and model weights after the rebuttal period.
>
> **W5: On the “Handcrafted” Pipeline**. Thank you for this comment. We agree that the current pipeline may appear somewhat handcrafted. However, for the fine-grained pixel-level drift studied here, existing automatic evaluators are still not reliable enough. We therefore use human annotations not as ad hoc heuristics, but as reliable supervision for capabilities that current automatic methods cannot yet capture. In our view, the key is not full automation itself, but a clear and principled evaluation framework.
>
> **W6: On Figure 1 Style.** Thank you for the suggestion. We agree that Fig. 1 can be more formal, and we will revise its emojis and overall style in the final version.

---

> > ### Author Rebuttal · Reviewer_BQbe · 2026-04-01
> >
> > Thanks to the authors, the response / rebuttal does address some of my concerns.
> >
> > Specifically remaining conerns:
> > 1. For W1 - I would really like to see the updated Fig. 2 in at least an anonymous link (similar for Fig 1. i.e., W6). It is always to show rather than to tell.
> > 2. W4: On Reproducibility - releasing code and dataset is really important, and I will follow up with the authors post the rebuttal period.
> > 3. Regarding the concern on **Soundness** - "The method section is not very well explained, and heavily references to the Appendix. For the method and key components atleast it should be self-sufficient in the main paper. For instance the Texture-Aware Enhancement Data part is not well explained." - this has still not been attended to or even acknowledged.
> > 4. For the figure that you show in https://anonymous.4open.science/r/anonymous-icml-2026-StableI2I/R2.pdf can the exact images in top 2 rows be shown with results from your proposed method ? That would make the results more credible.
> >
> > Rest I believe the response is ok. But the above have to be addressed.

---

> > > ### Author Response · Authors · 2026-04-02
> > >
> > > Thank you very much for the follow-up questions and for taking the time to further clarify the points you would like to see addressed. We sincerely appreciate your careful reading of our rebuttal and your constructive suggestions.
> > >
> > > **R1:** Thank you for your helpful suggestion. The figure corresponding to Fig. 2 is available at https://anonymous.4open.science/r/anonymous-icml-2026-StableI2I/R4.pdf, and the figure corresponding to Fig. 1 is available at https://anonymous.4open.science/r/anonymous-icml-2026-StableI2I/R5.pdf. Due to the limited space in the main paper, we were unable to further enlarge the figures. Instead, we have added as much relevant detail as possible and supplemented the explanation in the captions.
> > >
> > > **R2:** We have released our inference code and README at https://anonymous.4open.science/r/anonymous-icml-2026-StableI2I/code.zip. The environment setup is the same as that of Qwen3-VL, and the code can be loaded through the transformers library. It is worth noting that our model weights are 17 GB and the dataset is 168.6 GB, so we are unable to provide them through a single anonymous link. You may also use the official Qwen3-VL weights to test the inference code.
> > >
> > > **R3:** Thank you for pointing this out. We acknowledge that the explanation of the method, especially the Texture-Aware Enhancement Data component, is not sufficiently self-contained in the main paper, and our previous response does not address this concern clearly enough. This part is intended to illustrate the auxiliary augmentation data used to improve fine-grained perceptual sensitivity. However, due to the page limit, its presentation in the main paper is incomplete. For example, the figure for Texture-Aware Enhancement Data is actually part of the same large figure as Fig. 2, but we have to split it into two parts because of the space constraint, keeping only the main annotated data in the paper and moving the auxiliary augmentation data to the Appendix. This weakens the clarity of the overall explanation, and we sincerely apologize for this. In the final version, we will revise the paper structure and expand the description in the main paper so that the key method components, including Texture-Aware Enhancement Data, are explained more clearly and self-sufficiently.
> > >
> > > **R4:** Thank you for your question. We have provided the test outputs of our method in the figure available at https://anonymous.4open.science/r/anonymous-icml-2026-StableI2I/R6.pdf
> > >
> > > We are grateful for these valuable suggestions, which help us further improve the paper. We hope the clarifications above are helpful, and we will make these points more explicit in the final revision.

---

### Official Review · Reviewer_C18s · 2026-03-12

**Soundness:** 3
**Presentation:** 3
**Significance:** 2
**Originality:** 3
**Overall Recommendation:** 5
**Confidence:** 3

**Summary:**

The authors propose the StableI2I framework for validating image consistency after editing/restoration.

Update:
After discussion raised to accept

**Compliance With Llm Reviewing Policy:**

Affirmed.

**Key Questions For Authors:**

1. I strongly missed the code and parts of the dataset in the supplementary material. If these were provided, I would likely give a higher evaluation. Could the authors please share them via an anonymous storage repository?
2. Could you please clarify more precisely where the reference image is used and where it is not?

**Limitations:**

yes

**Strengths And Weaknesses:**

Strengths
* The authors annotate a large dataset to evaluate the image-to-image task.
* The authors propose a model that demonstrates strong performance in image editing assessment, which is an important component in many modern image processing tasks.


Weaknesses
* The paper lacks a detailed analysis and description of restoration scenarios. There exists a wide range of possible distortions, their severity levels, and different approaches to removing or correcting them. A clearer analysis would be important here, as the capabilities of different models on these tasks may vary significantly.
* Evaluating a wider range of open-source models could also be valuable for the analysis. In addition, simple image quality metrics, especially no-reference metrics, might perform well on certain aspects of the task (e.g., low-level distortions). Including such comparisons would be very informative for understanding their impact.
* The differences in the image editing examples are difficult to see. A possible solution would be to include the same images in higher resolution in the supplementary material.
* It is a minor weakness. The image on the 1st page is hard to understand, it would be very helpfull if it described the problem clearer.

---

> ### Author Rebuttal · Authors · 2026-03-28
>
> **W1: Clarification on restoration scenarios and degradation coverage.** Thank you for this helpful suggestion. We agree that restoration scenarios involve diverse degradation types, severity levels, and correction paradigms, so clear coverage is important. In the appendix (Sec. *A.1, Data Construction and Statistics*), we provide a detailed description of the degradation settings used in dataset construction. Specifically, following the Real-ESRGAN [1] degradation pipeline, we include salt-and-pepper noise, Gaussian noise, compression noise, Gaussian blur, directional blur, and mean blur, each with three severity levels. Table 5 further summarizes multiple open-source datasets covering diverse degradation types. In addition, we also detail the correction methods used in dataset construction, including unified generative models, restoration models, traditional GAN-based methods, and recent diffusion-based approaches.
>
> **W2: Baseline diversity and comparison with NR metrics.** Thank you for the suggestion. Regarding the first point, the generative models included in our evaluation already cover representative methods across different sampling paradigms and architectures. We also compared our method with conventional NR metrics through a dedicated pairwise preference study on ImgEdit. Specifically, we collected outputs from multiple generative models to construct 120 image pairs, and retained 112 valid pairs after filtering. Human annotators judged which output had higher fidelity, and a metric was counted as correct if its ranking agreed with human preference. StableI2I achieves 75.89% agreement, substantially outperforming CLIP-IQA (33.04%), MANIQA (39.29%), MUSIQ (35.71%), ArtiMuse (48.21%), and ImgEdit-Judge (32.69%).
>
> More importantly, conventional IQA methods generally measure the degradation level of a single image, whereas our evaluation focuses on whether the output image is degraded relative to the input image. Therefore, if the input is already degraded and the output faithfully preserves the same degradation, our method does not regard this as additional information loss, while conventional NR metrics may still assign a low quality score. This distinction is particularly important for image editing, where preserving the original visual information is often more critical than measuring absolute image quality. A concrete example is shown in: https://anonymous.4open.science/r/anonymous-icml-2026-StableI2I/R1.pdf
>
> **W3: Improving the visual clarity of qualitative examples.** Thank you for the helpful suggestion. We agree that some differences in the image editing examples may be difficult to see at the current resolution. In the final version, we will include higher-resolution versions of the same examples in the supplementary material, and where appropriate, add zoomed-in crops to highlight the edited regions more clearly.
>
> **W4: Clarifying the first-page figure.** We sincerely apologize for the confusion. What we show here is an image editing task: on the left are the input image and the instruction, and on the right are the results generated by two different models. Since image generation results generally do not have ground truth for direct quality evaluation, we can only rely on certain IQA and IAA metrics. The core issue is that these metrics cannot adequately use the original input image as a reference to determine whether content errors have occurred. In particular, ImgEdit-Judge corresponds to the fidelity dimension score, but its scoring procedure is overly coarse, which can also lead to biased results. Looking at the two outputs, although the GPT-generated result is indeed more visually appealing, regions other than the house and the trees have also been altered, for example through content repainting and information loss. This highlights the urgent need for a model like ours that can make more precise judgments.
>
> **Q1: Code and Dataset Release.** Thank you for the helpful suggestion. We agree that providing these resources would improve the evaluation. At this stage, we cannot release the full model weights anonymously, as the 8B checkpoint is too large for practical distribution during rebuttal. We will publicly release the model weights, pip package, and relevant datasets after the rebuttal process.
>
> **Q2: Clarification of Experimental Details.** Specifically, our focus is on I2I tasks, so there must be both an input image and an output image; the reference image is therefore the input image. Every task must refer to it, and this is also the core of our paper. For the exact insertion method, please see  *A.3.1, STABLEI2I-BENCH EVALUATION TEMPLATES*. In the phrase “The first image `<image>`”, the `<image>` token is replaced with the reference image.
>
> [1] Wang X, Xie L, Dong C, et al. *Real-ESRGAN: Training real-world blind super-resolution with pure synthetic data* \[C\]//Proceedings of the IEEE/CVF International Conference on Computer Vision. 2021: 1905–1914.

---

> > ### Author Rebuttal · Reviewer_C18s · 2026-03-31
> >
> > I thank the authors for the detailed rebuttal. My main concern remains the absence of code for verification; however, I acknowledge that it may be too large or otherwise impractical to share at this stage. I will therefore follow up on this after publication. I am ready to raise my verdict to accept.

---

> > > ### Author Response · Authors · 2026-04-01
> > >
> > > Thank you very much for your thoughtful follow-up and for your positive assessment. We truly appreciate your understanding regarding the practical difficulty of sharing the full codebase at this stage. We will make the code, model weights, and relevant resources publicly available after publication, and we would be very happy to follow up on any verification details then. Thank you again for your time and consideration.

---

### Decision · Program_Chairs · 2026-04-30

**Decision:**

Accept (regular)

**Comment:**

This paper addresses an important problem in image-to-image evaluation: measuring whether edited or restored images preserve the original content beyond instruction following and perceptual quality alone. Reviewers generally agreed that the paper is well motivated and that the proposed fidelity-centered formulation, together with the benchmark and evaluator, is a useful contribution.

The main strength of the submission is that it introduces a clearer evaluation target for this setting by explicitly separating semantic, structural, and low-level consistency. This gives the work practical value as an evaluation resource and may help improve how future I2I systems are assessed and compared.

At the same time, the contribution is primarily in evaluation design, benchmark construction, and system integration, rather than in a fundamentally new learning or modeling method. Some concerns also remained regarding clarity and completeness of presentation, as well as the overall maturity of parts of the framework. These concerns do not outweigh the paper’s strengths, but they do limit the paper’s ceiling.

Overall, based on the reviewer feedback and the practical value of the proposed evaluation perspective, I believe the paper makes a useful and relevant contribution to the area.